# Leukaemia exposure alters the transcriptional profile and function of BCR::ABL1 negative macrophages in the bone marrow niche

Amy Dawson[1,8], Martha M. Zarou [1,8], Bodhayan Prasad [1],
Joana Bittencourt-Silvestre[2], Désirée Zerbst[1], Ekaterini Himonas [1],
Ya-Ching Hsieh[3], Isabel van Loon[1], Giovanny Rodriguez Blanco[3],
Angela Ianniciello[1], Zsombor Kerekes[1], Vaidehi Krishnan [4], Puneet Agarwal [5],
Hassan Almasoudi[1,6], Laura McCluskey[2], Lisa E. M. Hopcroft [2], Mary T. Scott [1],
Pablo Baquero[1,7], Karen Dunn [2], David Vetrie [1], Mhairi Copland [2],
Ravi Bhatia [5], Seth B. Coffelt [1,3], Ong Sin Tiong [4], Helen Wheadon [2],
Sara Zanivan[1,3], Kristina Kirschner [1,3] ✉ & G. Vignir Helgason [1] ✉

Macrophages are fundamental cells of the innate immune system that support normal haematopoiesis and play roles in both anti-cancer immunity and tumour progression. Here we use a chimeric mouse model of chronic myeloid leukaemia (CML) and human bone marrow (BM) derived macrophages to study the impact of the dysregulated BM microenvironment on bystander macrophages. Utilising single-cell RNA sequencing (scRNA-seq) of Philadelphia chromosome (Ph) negative macrophages we reveal unique subpopulations of immature macrophages residing in the CML BM microenvironment. CML exposed macrophages separate from their normal counterparts by reduced expression of the surface marker CD36, which significantly reduces clearance of apoptotic cells. We uncover aberrant production of CML-secreted factors, including the immune modulatory protein lactotransferrin (LTF), that suppresses efferocytosis, phagocytosis, and CD36 surface expression in BM macrophages, indicating that the elevated secretion of LTF is, at least partially responsible for the supressed clearance function of Ph- macrophages.

Haematopoiesis is a tightly controlled process that gives rise to the entire blood system. In adult mammals, haematopoiesis primarily occurs in the long bones, where haematopoietic stem cells (HSCs) reside in specific niches that regulate their function and survival[1]. The bone marrow (BM) niche and HSCs closely interact to maintain HSC quiescence, self-renewal and differentiation[1,2]. This relationship requires communication of HSCs with the surrounding microenvironment via secretory factors and direct cell-cell contact with many cell populations[1]. One such population includes resident macrophages that play a pivotal role in HSC maintenance[3], self-renewal[4] and dormancy[5]. Nitric oxide and spermidine release by BM macrophages controls the self-renewal of HSCs[6] and DARC+ macrophages control long-term HSC (LT-HSC) dormancy via CD82/DARC mediated TGFβ/Smad3 signalling[5]. Additionally, the loss of macrophages results in mobilisation of HSCs into the blood, via the loss of CXCL12 from stromal cells[3]. However, in leukaemia, the niche is disrupted to protect

the survival of leukaemic stem cells (LSCs) at the expense of normal HSCs[7,8], and little is known about the effect of these niche modifications on BM macrophage populations and the mechanisms involved.

Chronic myeloid leukaemia (CML) is a haematological malignancy characterised by a reciprocal translocation between chromosome 9 and 22 to give rise to the oncogene *BCR::ABL1*[9,10]. CML is a classical stem cell-driven cancer, with BCR::ABL1 translocation arising in an HSC, and the transformed LSC driving a myeloproliferative phenotype through constitutive kinase activity. The CML LSC requires a modified niche to support LSC survival and function. Increased levels of pro-inflammatory cytokines such as IL-6[11] have been measured within the niche and shown to drive normal HSC cycling and acquisition of genetic signatures similar to the LSC counterpart[11,12]. However, it is becoming increasingly evident that bystander cells in the niche are also modified to create the leukaemic-supportive environment. Recent evidence has demonstrated an increased infiltrate of CD68+p-STAT1⁻cMAF+ macrophages in CML BM biopsies compared to control samples[13]. This population of macrophages is associated with immunosuppression[13], perhaps suggesting macrophages are contributing to a CML-supportive environment. Interestingly, Bührer et al. demonstrated that red pulp macrophages contribute to disease progression and therapy resistance by maintaining LCS quiescence[14]. However, it should be noted that macrophages are highly plastic cells, with the capacity to play both pro and anti-tumourigenic roles as strongly evidenced in solid tumours[15,16].

In this study, we utilise a state-of-the-art chimeric CML mouse model to demonstrate that the CML environment promotes myelopoiesis of normal HSCs, with an increase in neutrophils, monocytes and macrophages in the BM. Further analysis of these CD11b+F4/80+BCR::ABL1⁻ BM macrophages by scRNA-seq shows that macrophage populations within the CML BM are altered both functionally and transcriptionally by CML exposure. Here we demonstrate that CML exposure promotes unique macrophage subpopulations not found in the normal BM. CML-exposed macrophages are largely associated with more immature signatures when exposed to CML. Most notably, we establish that exposure to the CML microenvironment results in downregulation of cell surface CD36 expression, leading to a functional impairment of macrophage efferocytosis. Furthermore, we demonstrate that c-Kit+ CML stem and progenitor cell populations are actively modifying the BM niche through dysregulated protein secretion. We uncover that the secreted factor LTF is elevated in CML and contributes to the observed suppression of CD36 levels and the reduction in macrophage phagocytosis and efferocytosis.

## Results

### CML promotes myelopoiesis at the expense of lymphoid cells

To initially investigate the effect of BCR::ABL1 activity on normal haematopoietic cells, an inducible chimeric CML model was established. In a setup similar to the model generated by Welner and colleagues[11] we established a competitive transplant model exploiting the expression of the two alleles of the common antigen CD45. C57/Bl6-CD45.1 (WT) BM was combined with either SCLtTA × BCR::ABL1⁻-CD45.2 (Control; no oncogene) or SCLtTA × BCR::ABL1+-CD45.2 (CML) BM and transplanted into sublethal irradiated C57/Bl6 WT mice (Fig. 1a). Chimerism was established and both CD45.1 and CD45.2 cells were detected in the blood and BM of mice (Supplementary Fig. 1a–c). Following the removal of tetracycline for 15 days, a CML-like disease was established, with an increase in Gr-1+CD11b+ cells in the blood and BM of mice transplanted with WT:CML BM (Fig. 1b, c; Supplementary Fig. 1d).

Analysis of the BM upon sacrifice revealed that CML induction increases the percentage of CD45.1 Lin⁻c-Kit+ (LK) progenitor cells compared to control but had no significant effect on the frequency of Lin⁻Sca-1+c-Kit+ (LSK) cells or more primitive LT-HSCs (Supplementary Fig. 2a–d). Furthermore, analysis of the cellular composition of CML

BM revealed that in addition to BCR::ABL1+ driven reduction in CD19+ B cell frequency, CD45.1+CD19+ B cells (CML exposed) were also significantly reduced in frequency (Fig. 1d; Supplementary Fig. 2e, f) and in absolute cell number (Supplementary Fig. S3a). Additionally, we found that CD8+ and CD4+ T cells were largely unaffected in the CD45.2 arms, but CD8+ T cells were significantly reduced in non-transformed CD45.1 BM (Fig. 1e, f; Supplementary Fig. 2g, h and 3b, c). Interestingly, in addition to a reduction in normal B and CD8+ T cells, we observed an increase in myeloid cells in comparison to control. We observed that CML exposure results in a significant increase in the frequency of BCR::ABL1⁻ neutrophils, macrophages and monocytes (Fig. 1g–i; Supplementary Fig. 2i–m), and a rise in the absolute cell number of both macrophages and monocytes (Supplementary Fig. 3d–f). These results demonstrate that our chimeric CML mouse model alters the balance of normal haematopoietic cells towards a myeloid bias, thus supporting that the CML niche supports myelopoiesis of the BCR::ABL1⁻ population.

### CML exposure alters macrophage gene expression profiles

Recent reports in the field of acute myeloid leukaemia (AML) show that depletion of BM phagocytes results in an increase in tumour burden in a mouse model of AML[17]. Given the importance of macrophages in the healthy BM niche, and their potential role in controlling leukaemogenesis, we investigated the importance of macrophages in the CML BM microenvironment. Firstly, through anti-CSF1R mediated macrophage reduction (Fig. 2a, b), we observed that animals with a decreased pool of BM macrophages had significantly reduced survival when compared to leukaemic animals treated with an isotype control (Fig. 2c), which correlated with an increase in Gr-1+CD11b+ cells in the blood at experimental endpoint (Fig. 2d, e). These findings indicate that macrophages play a critical role in controlling leukaemia progression and burden in CML.

Given that macrophage gene expression patterns are intrinsically linked to their functional status, we investigated the impact of CML exposure on the macrophage transcriptome as first step to understanding the interplay between the CML microenvironment and BM macrophages. Here we demonstrate that CML-exposed CD45.1+CD11b+F4/80+ macrophages, sorted directly from the leukaemic mouse BM, are altered transcriptionally compared to control macrophages, isolated from non-induced BM (Fig. 3a, b). Interestingly, *Cd36* and *Lamp2*, were significantly downregulated in macrophages exposed to a CML niche (Fig. 3b). Both *Cd36* and *Lamp2* are known to be associated with many functions, including, to our interest, phagocytosis. Downregulation of *Cd36* and *Lamp2* in CML-exposed macrophages was also evident when compared with Ph+ macrophages (Supplementary Fig. 4a, b and Supplementary Tables 1 and 2). Overall, this suggests that a unique population of macrophages exists in the oncogene-negative fraction on the CML BM. Furthermore, we confirmed that *BCR::ABL1* expression was restricted to the off tetracycline CD45.2 BM fraction and not expressed in the CD45.1 BM fraction (Supplementary Fig. 4c).

### scRNA-seq reveals unique macrophage populations in the CML niche

The results thus far indicate that at a population level, the CML environment alters the transcriptional phenotype of BM macrophages. However, macrophages are notoriously plastic cells that can react heterogeneously to a range of environmental stimuli. To investigate macrophage heterogeneity in the CML niche, we performed scRNA-seq analysis on CD45.1+CD11b+F4/80+ macrophages isolated from the BM of control and CML chimeric mice. Unbiased clustering algorithms categorised macrophages into 6 different subpopulations (Fig. 3c). Clustering analysis revealed populations found almost exclusively in macrophages isolated from control mice (C1 & C5), which were found to separate independently from C2 and C6, mainly containing

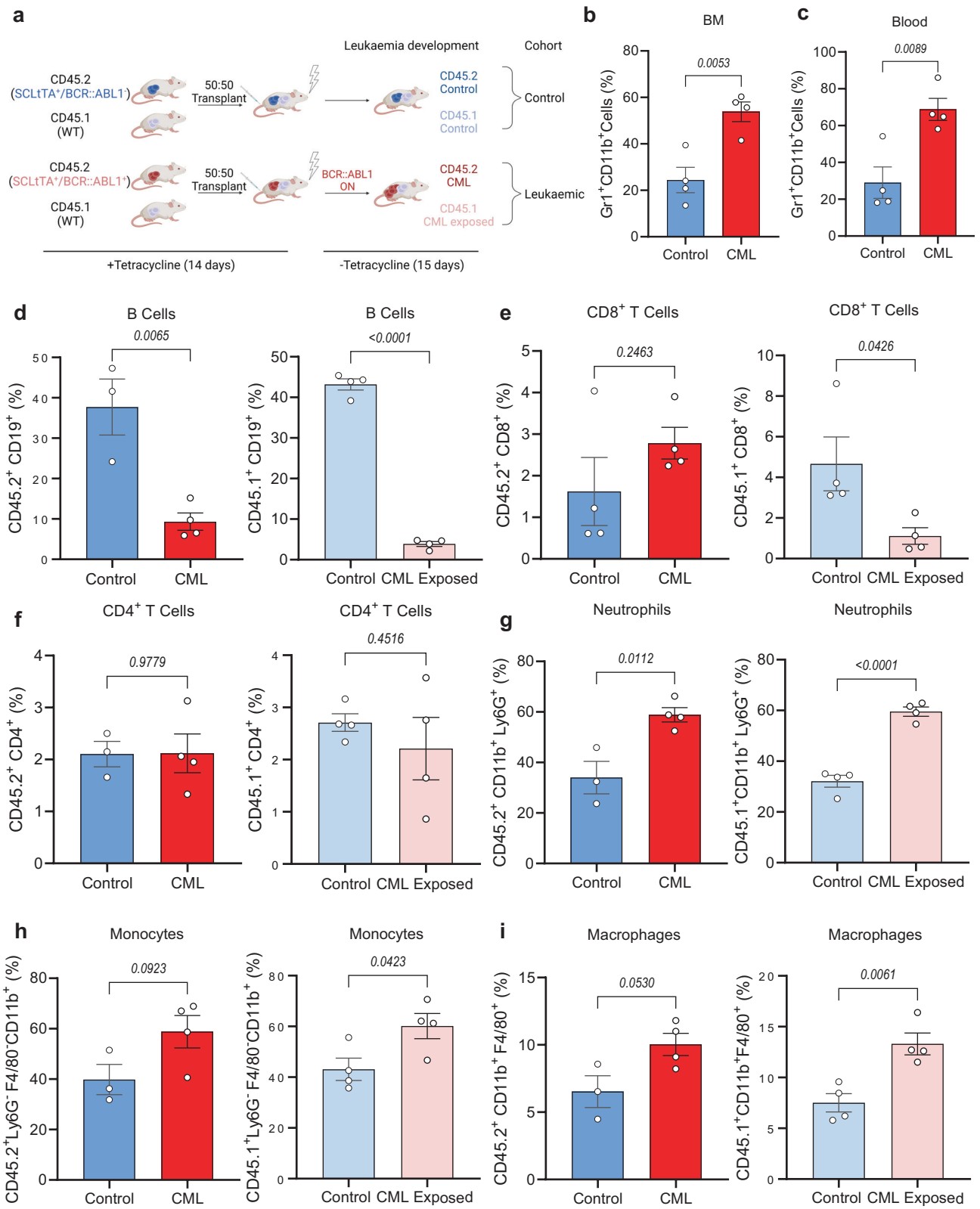

macrophages isolated from the CML-exposed environment (Fig. 3c). A further two clusters were found to contain a mixture of macrophages from both the control and CML-exposed environment (C3 & C4; Fig. 3c). Quality control measures were performed to ascertain that there was no plate-specific effect driving the observed cell clustering (Supplementary Fig. 5a, b). Overall, these results confirm our bulk analysis that BM macrophages are transcriptionally altered upon

exposure to the CML niche. However, by employing scRNA-seq we have uncovered further subpopulations of macrophages in the CML niche.

To further characterise each identified macrophage population, the top marker genes driving unsupervised clustering were identified (Fig. 3d). CML-exposed macrophages found in C2 and C6 were associated with genes related to alternatively activated macrophages,

**Fig. 1 | CML results in an increase in Ph⁻ progenitor cells and myeloid skew at the expense of lymphocytes. a** Schematic outline of experimental design to generate CML chimeric mice. $7.5 \times 10^5$ bone marrow (BM) cells from either CD45.2 SCLtTA⁺/BCR::ABL1⁻ (Control) or SCLtTA⁺/BCR::ABL1⁺ (CML) mice mixed with $7.5 \times 10^5$ CD45.1 BM cells from wild-type (WT) mice was transplanted into WT mice. CML-exposed cells refer to CD45.1 WT BM cells exposed to CD45.2 SCLtTA⁺/BCR::ABL1⁺ (CML) BM cells. CML-exposed cells have been compared to CD45.1 WT BM cells exposed to CD45.2 SCLtTA⁺/BCR::ABL1⁻ (Control). Created with BioRender.com (Agreement number: XA268HUG33). **b** Quantification of myelo-proliferation in BM (**b**) and blood (**c**) at experimental endpoint ($n = 4$ mice per experimental arm) **d–i** Flow cytometry analysis of CD45.1 and CD45.2 arms of BM of chimeric mice after 15 days off TET. Percentage of CD19⁺ B cells ($n = 3$ mice for CD45.2⁺CD19⁺ control, $n = 4$ mice for CD45.2⁺CD19⁺ CML, CD45.1⁺CD19⁺ control and CD45.1⁺CD19⁺ CML exposed) (**d**), CD8⁺ T cells ($n = 4$ mice per experimental arm) (**e**), CD4⁺ T cells ($n = 3$ mice for CD45.2⁺CD4⁺ control, $n = 4$ mice for CD45.2⁺CD4⁺ CML, CD45.1⁺CD4⁺ control and CD45.1⁺CD4⁺ CML exposed) (**f**), CD11b⁺Ly6G⁺ cells ($n = 3$ mice for CD45.2⁺CD11b⁺Ly6G⁺ control, $n = 4$ mice for CD45.2⁺CD11b⁺Ly6G⁺ CML, CD45.1⁺CD11b⁺Ly6G⁺ control and CD45.1⁺ CD11b⁺Ly6G⁺ CML exposed) (**g**), CD11b⁺F/80⁺ cells ($n = 3$ mice for CD45.2⁺CD11b⁺F/80⁺ control, $n = 4$ mice for CD45.2⁺CD11b⁺F/80⁺ CML, CD45.1⁺CD11b⁺F/80⁺ control and CD45.1⁺CD11b⁺F/80⁺ CML exposed) (**h**), and CD11b⁺Ly6G⁻F/80⁻ cells ($n = 3$ mice for CD45.2⁺CD11b⁺Ly6G⁻F/80⁻ control, $n = 4$ mice for CD45.2⁺CD11b⁺Ly6G⁻F/80⁻ CML, CD45.1⁺CD11b⁺Ly6G⁻F/80⁻ control and CD45.1⁺CD11b⁺Ly6G⁻F/80⁻ CML exposed) (**i**). Data are shown as the mean ± s.e.m. $P$-values were calculated using unpaired two-tailed $t$-test (**b–i**). Source data are provided as a Source Data file.

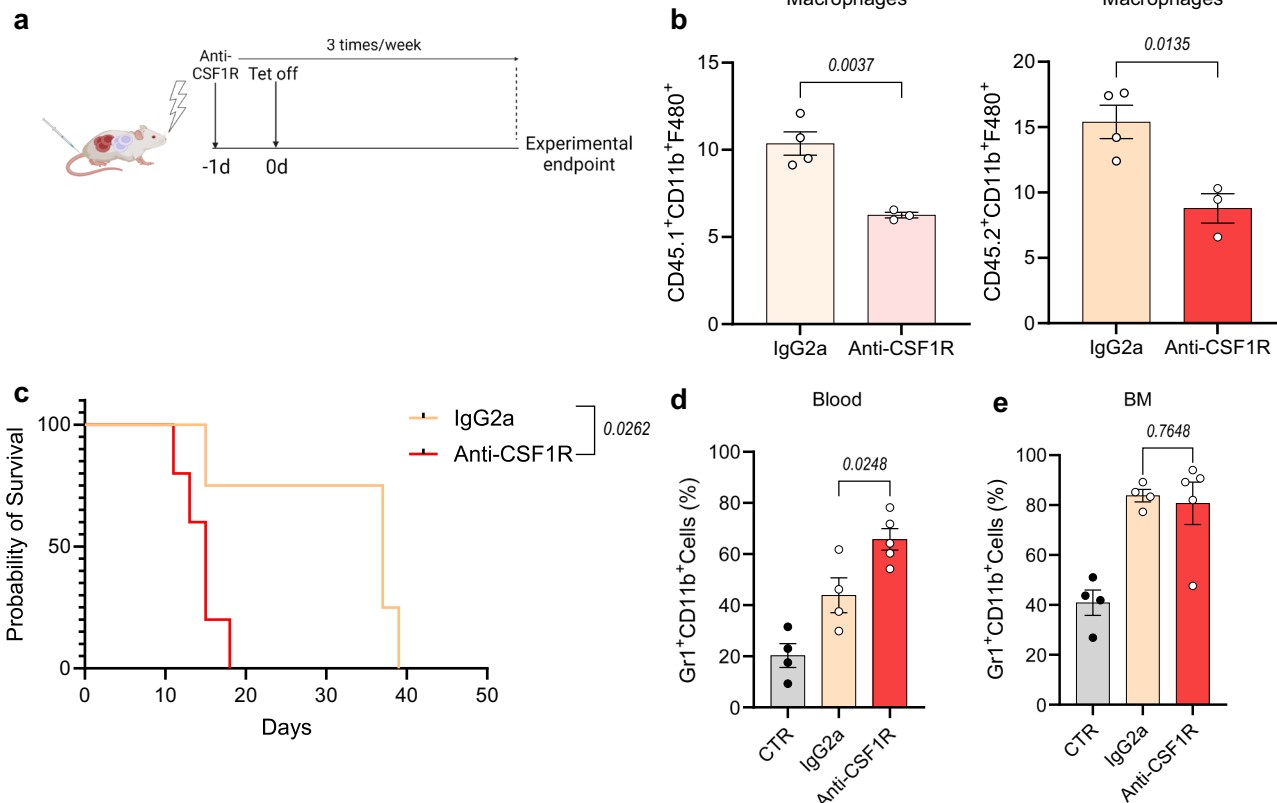

**Fig. 2 | Macrophage depletion promotes leukaemogenesis. a** Schematic overview of macrophage depletion following CSF1R blockage in CML chimeric mice. Mice were treated with anti-CSF1R antibody or IgG2a isotype control one day prior to tetracycline withdrawal. Anti-CSF1R was then administered 3 times per week. Schematic created with BioRender.com (Agreement number: FA268HUUTO). **b** Percentage of CD11b⁺F/80⁺ cells in CD45.1 or CD45.2 BM following treatment with anti-CSF1R ($n = 3$ mice) or IgG2a isotype control ($n = 4$ mice). **c–e** Overall survival (**c**) and myeloproliferation in blood (**d**) and BM (**e**) at experimental endpoint following treatment with anti-CSF1R ($n = 5$ mice) or IgG2a isotype control ($n = 4$ mice). The control (CTR) experimental arm ($n = 3$ mice) refers to irradiated control mice. Data are shown as the mean ± s.e.m. $P$-values were calculated using unpaired two-tailed $t$-test (**b, d, e**), or log-rank (Mantel–Cox) test for survival analysis (**c**). Source data are provided as a Source Data file.

including enrichment for genes such as *Tgfbi*, *Lgals1* and *Ly6c2* (Fig. 3d; Supplementary Fig. 6a). Interestingly, we also noted that C1, a control population, had high expression of *Cd36* that appeared to be low in our CML-exposed macrophages (Fig. 3d, e; Supplementary Fig. 6b). *Nr4a1* expression, previously implicated in phagocyte survival and phagocytosis dependent upregulation of mertk[18], was also enriched in C1 (Fig. 3d, e, Supplementary Fig. 6b). Interestingly, we had previously identified significantly lower *Cd36* mRNA expression in bulk CD45.1⁺ CD11b⁺F/80⁺ macrophages exposed to CML (Fig. 3b; Supplementary Table 1). A candidate marker analysis performed using COMET[19], revealed that CD36 is ranked as a candidate marker for isolation of cluster C1 as CD36⁺ (Supplementary Fig. 6c), therefore we propose that CD36 can be used as a candidate marker to isolate

CD45.1⁺ CD11b⁺ F/80⁺ CD36⁻ macrophages as those functionally altered by the CML niche. CML-exposed subpopulations C2 and C6 also display a gene signature with high expression of immature macrophage genes such as *Ly6c2* and an anti-inflammatory signature including enrichment for *Tgfbi* and *Lgals1* (Fig. 3d; Supplementary Fig. 6a). C5 (a control population) also appears to represent an immature population of macrophages present in the normal BM niche, with high expression of *Ly6c2* and *S100a6* (Fig. 3d; Supplementary Fig. 6a, b). Finally, COMET analysis revealed that the C2 population has an increased surface expression of *Lgals1* and *Cd14*, confirming an immature and anti-inflammatory signature for CML-exposed macrophages (CML-exposed cluster; Supplementary Fig. 6d, e).

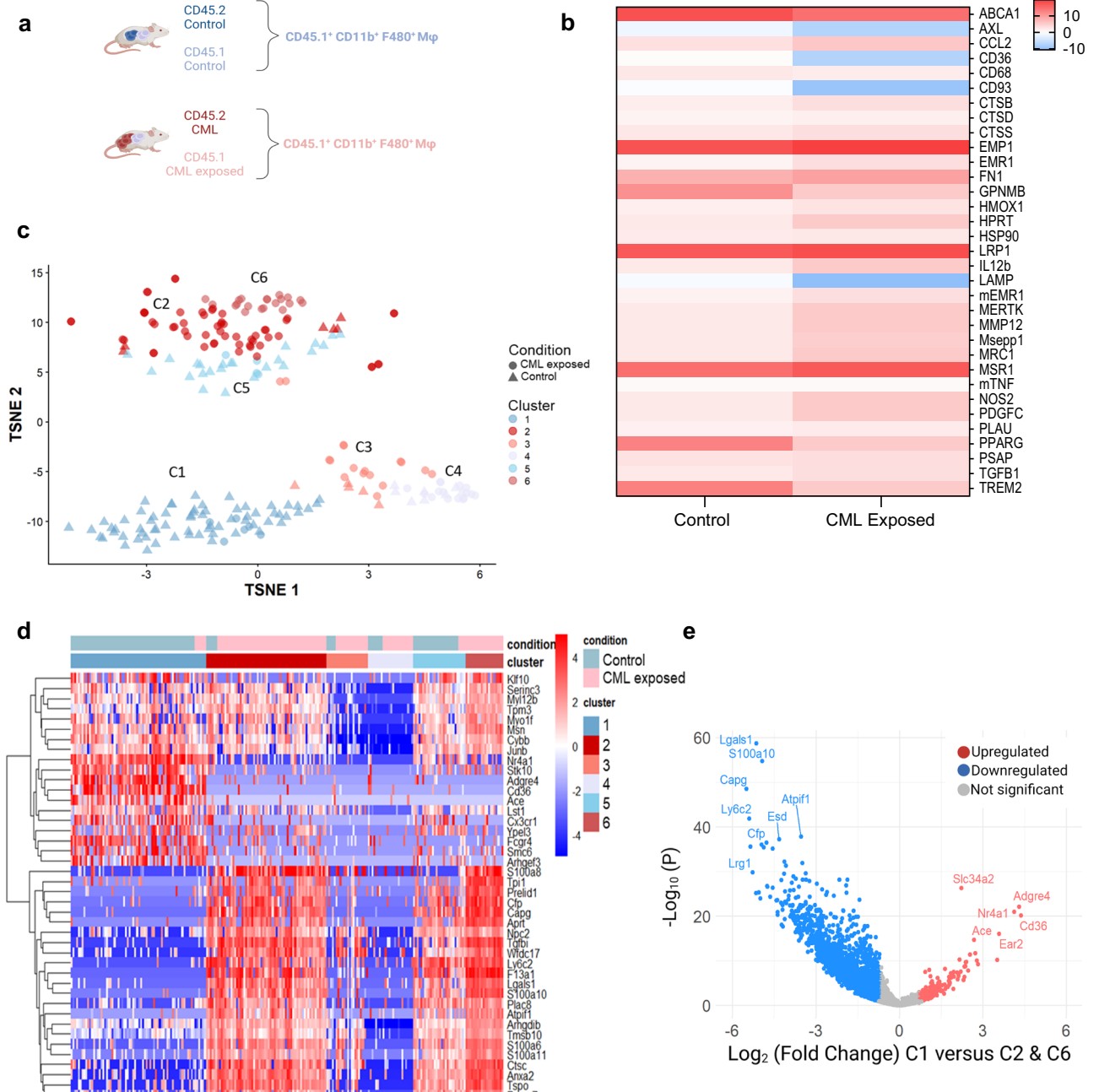

**Fig. 3 | CML-exposed macrophages have altered gene expression and form unique subpopulations. a, b** Schematic outline of experimental design (**a**) and heatmap (**b**) of Fluidigm gene expression analysis of CD11b⁺F4/80⁺ macrophages sorted from CD45.1 (Control/CML exposed) BM (*n* = 7 mice for CD11b⁺F4/80⁺ macrophages exposed to control; *n* = 4 mice for CD11b⁺F4/80⁺ CML-exposed macrophages). Schematic created with BioRender.com (Agreement number: MH268HV0JI). **c** t-Distributed Stochastic Neighbour Embedding (t-SNE) visualisation of CD45.1⁺ CD11b⁺F4/80⁺ cells from control and CML BM scRNA-seq. **d** Unsupervised clustering of top 20 marker gene expression heatmap. **e** Volcano plot of differentially expressed genes in cluster 1 (C1) compared to cluster 2 (C2) and 6 (C6). Upregulated genes with a C1 versus C2 and C6 log2 (fold change) of at least 0.7 are shown in red, while downregulated gene with a C1 versus C2 and C6 log2 (fold change) of at least −0.7 are shown in blue. Differential expression (DE) testing was carried out using the default wrapper function present inside the scran package in R (DeSeq2), which combines pairwise two-tailed *t*-test into a single rank list markers. The Benjamini−Hochberg method was used to correct for multiple comparisons (**e**). Source data are provided as a Source Data file.

To further delineate the impact of the CML microenvironment on subpopulations of BM niche macrophages, pathway analysis was performed on each macrophage cluster. Gene set enrichment analysis (GSEA) further highlights that macrophages exposed to a CML niche have differential enrichment for inflammatory signalling pathways compared to control. Analysis of single-cell clusters demonstrated that TNF signalling via NF-κB, is significantly downregulated in CML-exposed macrophages found in C2 (*FDR q = 0.046, NES = −1.59*) and C6

(*C6: FDR q = 0.044, NES = −1.56*; Supplementary Fig. 7a, b). In contrast, in the control cluster C1, a significant positive enrichment for TNF signalling via NF-κB (*FDR q = 0.00, NES = 2.18*) was observed (Supplementary Fig. 7c). To support these findings, BM-derived macrophages (BMDM) generated from WT C57/Bl6 mice, were conditioned with medium from mouse control (conditional medium; CM) or CML c-Kit⁺ enriched BM (leukaemia conditional medium; LCM). Treatment with LCM suppressed the expression of TNF in macrophages although no

changes in phosphorylation pattern of the p65 subunit of NF-κB were observed (Supplementary Fig. 7d, e). Furthermore, GSEA revealed that CML-exposed macrophages appear to be less metabolically active than their WT counterparts. C2 and C6 macrophages had reduced overall expression gene sets associated with oxidative phosphorylation (*FDR q = 0.173, NES = −1.47; FDR q = 0.35 NES = − 1.43*, respectively) (Supplementary Fig. 8a, b). To validate this observation, c-Kit[+] cells isolated from the mouse BM of CML, and control mice, were co-cultured with WT macrophages and the metabolic activity of the macrophages assessed after 48 h culture. Macrophages co-cultured with CML c-Kit[+] cells were observed to have a reduced activity in central carbon metabolism pathways (oxidative phosphorylation and glycolysis) compared to macrophages co-cultured with WT c-Kit[+] cells (Supplementary Fig. 8c–e).

Lastly, to investigate the heterogeneity of human BM macrophages following CML development, we applied a scRNA-seq dataset of human BM mononuclear cells isolated from either CML patients at diagnosis or healthy counterparts[20]. Unbiased clustering analysis categorised human BM mononuclear cells into 16 different subpopulations (Supplementary Fig. 9a, b). The previously named myeloid cluster (C7) was identified as a macrophage-enriched subpopulation based on cluster-specific expression of known macrophage markers CD68, CSF1R, SIGLEC1 and CD163 (Supplementary Fig. 9a, b). Within the myeloid cluster, subcluster 5 displayed the highest expression of CD68 and CSF1R and was classified as a macrophage subcluster (Supplementary Fig. 9c, d). Further clustering analysis on the macrophage subpopulation demonstrated that CML BM macrophages separate independently and display a strong inflammatory signature when compared with macrophages deriving from healthy individuals (Supplementary Fig. 9e, f). Given that most of CML BM macrophages at diagnosis derive from malignant LSCs and are therefore Ph[+21], the observed inflammatory signature agrees with our findings that Ph[−] macrophages are transcriptionally different from Ph[+] macrophages (Supplementary Fig. 4b; Supplementary Table 2).

Overall, we demonstrate that the CML BM niche drives high expression of genes related to alternatively activated macrophages such as *Tgfbi* and *Lgals1*. We identified that both subpopulations of CML-exposed macrophages harbour an immature transcriptional signature, which appears to cluster with an immature subpopulation found under homoeostatic conditions. Furthermore, CML-exposed macrophages undergo a metabolic switch characteristic of immunosuppressive macrophages as compared to control macrophage clusters. This suggests the CML environment favours immature, anti-inflammatory macrophage populations.

## CML exposure suppresses clearance of apoptotic cells through downregulation of CD36

We next sought to investigate the impact of suppressed CD36 expression on BM macrophage function. CD36 is a well-characterised phagocytic receptor, known to be essential for macrophage-mediated clearance of apoptotic cells[22]. To explore whether CML-driven suppression of CD36 has an effect on the clearance capacity of BM macrophages, we co-cultured WT BMDM with fluorescent-labelled (Cell Trace Violet positive; CTV[+]) c-Kit[+] cells. This revealed a significant reduction in a CTV[+] signal in macrophages co-cultured with CTV[+] mouse CML stem/progenitor cells, in agreement with our transcriptional observation that CML exposure reduces phagocytosis gene expression levels (Fig. 4a, Supplementary Fig. 10a). To further assess clearance of apoptotic cells (efferocytosis), BMDM conditioned with LCM or CM were incubated with apoptotic K562 CML cells (with increased phosphatidylserine levels on their outer plasma membrane; Supplementary Fig. 10b). Mouse BMDM treated with LCM displayed a significant reduction in their efferocytosis capacity (Fig. 4b; Supplementary Fig. 10c). Additionally, BMDM exposed to LCM demonstrated reduced phagocytosis of latex beads (Fig. 4c), further supporting our

findings that CML exposure significantly affects the ability of BMDM to phagocytose and clear apoptotic cells.

As we observed that both co-cultured macrophages and those treated with CM resulted in alterations of phagocytic function, we next sought to identify the effect of CML-secreted factors on CD36 expression. Notably, macrophage exposure to LCM significantly reduces cell surface CD36 expression (Fig. 4d). Next, BMDM were treated with CM in the presence of the irreversible CD36 inhibitor sulfo-N-succinimidyl oleate (SSO). SSO exposure resulted in a significant reduction in efferocytosis, without any effect on clearance of latex beads (Fig. 4e, f). Lastly, BMDM treated with SSO had significantly increased expression of CD86 and CD301, similarly to macrophages exposed to LCM, whereas no effect was observed on CD206 expression (Fig. 4g–i).

## CML stem/progenitor cells have a dysregulated protein secretome

Thus far our findings demonstrated that CML-exposed macrophages are transcriptionally and functionally distinct from normal counterparts both at a population level (Fig. 3b) and at single-cell resolution (Fig. 3c–e). Additionally, we noted that LCM exposure can alter macrophage phenotype, promoting downregulation of CD36 expression, reduced phagocytic clearance of latex beads and efferocytosis (Fig. 4b–f). Therefore, we next sought to determine if CML-secreted factors were in part driving the observed changes in macrophages exposed to a CML environment. Firstly, we investigated whether primitive CML cells had a dysregulated secretome compared to control by performing bulk RNA-seq analysis on LT-HSC. LSK Flt3[−]CD150[+]CD48[−] cells were isolated from age and sex-matched controls and SCLtTA[+]/BCR::ABL1[+] mice (Fig. 5a). Gene Ontology (GO)-enrichment analysis demonstrated a significant downregulation of pathways related to cell adhesion (Supplementary Fig. 11) and upregulation of cell migration pathways (Fig. 5b). However, interestingly, we noted that both cytokine secretion and the positive regulation of cytokine production were amongst the top 20 enriched pathways in CML (Fig. 5b). To validate this in silico observation, we performed an unbiased mass spectrometry (MS)-based proteomic analysis of the culture medium obtained from BM-derived c-Kit[+] cells, to identify secreted proteins. A CML-like disease was induced in SCLtTA[+]/BCR::ABL1[+] mice by the removal of tetracycline for 15 days (Fig. 5c). BM c-Kit[+] cells were isolated from both CML and control mice and used to condition medium for 24 h, which was subsequently analysed by MS. This analysis revealed that LCM contained a significantly different secretory proteome compared to CM (Fig. 5d), with principal component analysis (PCA) of all detected proteins revealing a noticeable separation between control mice and CML mice (Supplementary Fig. 12). Following statistical analysis, 20 secreted proteins were found to be significantly dysregulated in CML compared to control c-Kit[+] cells (Fig. 5e). To our interest, we noted that the immune modulatory protein lactotransferrin (LTF) was modestly, yet significantly, increased in LCM (Fig. 5e). Furthermore, bulk RNA-seq analysis revealed that out of the 20 significantly changed proteins in LCM, only LTF and PLD1 were significantly differentially expressed by CML LT-HSC compared to control counterparts (Supplementary Fig. 13). Lastly, ELISA analysis of c-Kit[+] conditioned medium from both control and CML mice further confirmed that CML conditioned medium contained a significantly higher concentration of LTF than control (Fig. 5f).

## LTF exposure suppresses efferocytosis, phagocytosis and CD36 expression in mouse macrophages

To determine if LTF was contributing to the observed changes to macrophage function, BMDM were treated with CM, spiked with endotoxin-free bovine LTF. Strikingly, supplementation with LTF significantly decreased both the efferocytosis and phagocytosis capacity of mouse BMDM (Fig. 6a, b). We also observed that macrophages

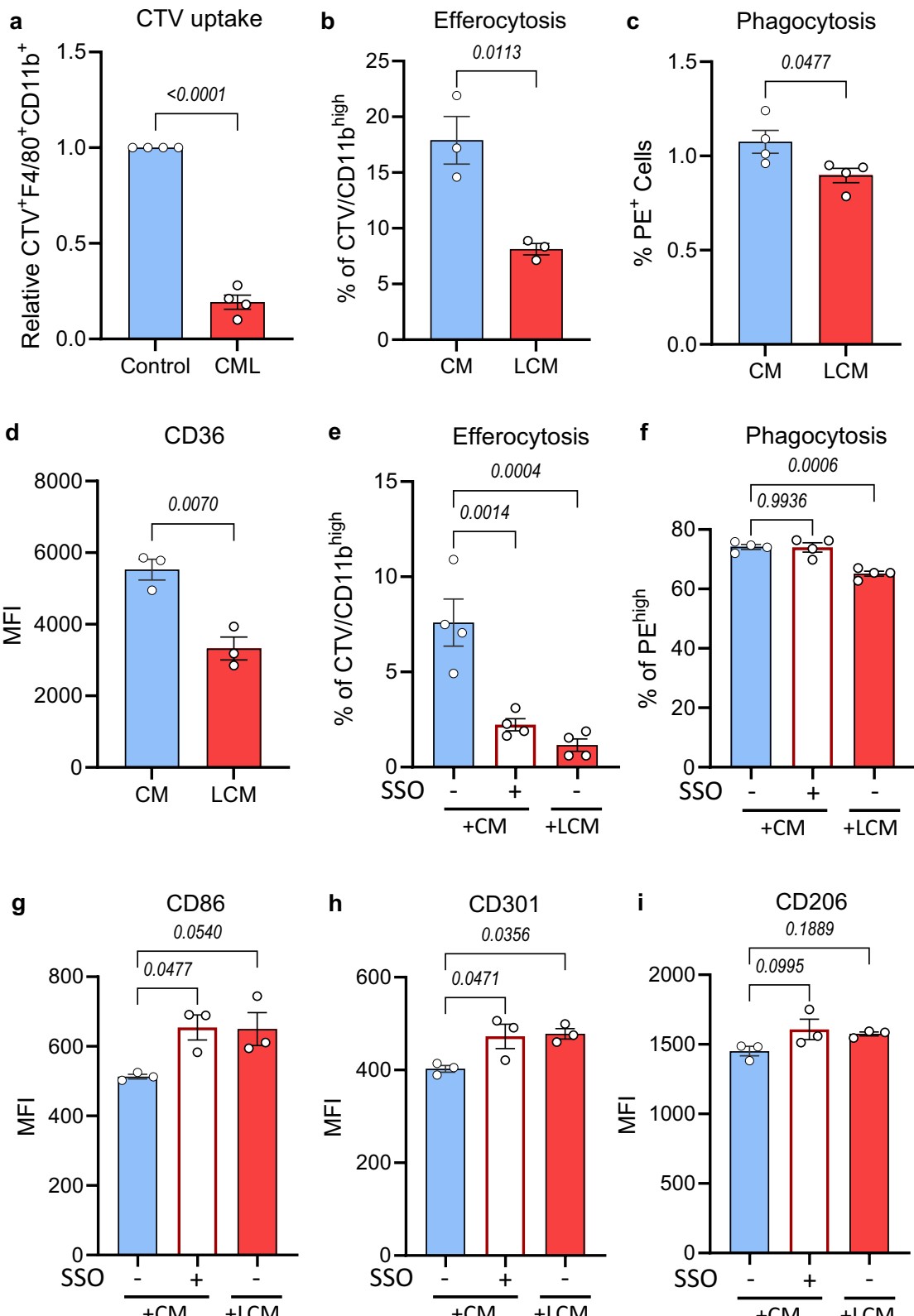

cultured with LTF had significantly increased expression of CD86, CD301, CD206 and MHC II, in a manner similar to that of macrophages treated with LCM (Fig. 6c–f). LTF treatment also suppressed the expression levels of *CD36* and increased the expression of a number of genes including *IL-12β*, *Nos2* and *Arg1* (Fig. 6g). CD36 cell surface expression was also suppressed by LTF exposure (Fig. 6h). While treatment of BMDM with LTF in regular growth medium had

comparable results to supplementing LTF into CM, we observed an opposite effect for phagocytosis of latex beads following treatment with LTF (Supplementary Fig. 14a, b). Of note, we observed decreased phosphorylation of the p65 subunit of NF-κB, p38 MAPK and p44/42 MAPK (ERK1/2) in BMDM exposed to LTF (Supplementary Fig. 14h). These results are in line with previous studies that have demonstrated that LTF has an anti-inflammatory effect by suppressing

**Fig. 4 | CML exposure suppresses clearance of apoptotic cells through down-regulation of CD36. a** CTV$^+$CD11b$^+$F4/80$^+$ flow cytometry analysis following Control/CML CTV$^+$ c-Kit$^+$ cells co-culture with BMDM for 48 h ($n = 4$ independent experiments). **b** Percentage of CTV/CD11b$^{high}$ BMDM conditioned with medium from control (CM) or CML c-Kit+ enriched BM (LCM) for 24 h in the presence of CML apoptotic cells for the final two hours ($n = 3$ independent experiments). **c** Quantification of phagocytosis of latex beads measured as percentage of PE$^{high}$ in BMDM following exposure to CM or LCM for 24 h ($n = 4$ independent experiments). **d** Surface marker expression of CD36 in BMDM measured as mean fluorescent intensity (MFI) following exposure to CM or LCM ($n = 3$ independent experiments). **e** Quantification of efferocytosis in BMDM conditioned with CM, CM supplemented with 50 μM sulfo-N-succinimidyl oleate (SSO), or LCM for 24 h ($n = 4$ independent experiments). **f** Quantification of phagocytosis of latex beads in BMDM following exposure to CM in the presence or absence of 50 μM SSO or LCM for 24 h ($n = 4$ independent experiments). **g**–**i** Surface marker expression of CD86 (**g**), CD301 (**h**) and CD206 (**i**) in BMDM exposed to CM with or without 50 μM SSO addition or LCM for 24 h ($n = 3$ independent experiments). Data are presented as the mean ± s.e.m. $P$-values were calculated using unpaired two-tailed $t$-test (**a**–**d**), or ordinary one-way ANOVA with Dunnet's multiple comparisons test (**e**–**i**). Source data are provided as a Source Data file.

lipopolysaccharide (LPS)-induced cytokine production[23]. Lastly, both basal and maximal mitochondrial respiration was decreased in BMDM following LTF treatment (Supplementary Fig. 14i–k). These results suggest that CML-derived LTF plays a role in the suppression of a pro-inflammatory response and CD36 expression in BM niche macrophages, ultimately reducing their clearance capacity.

### LTF exposure alters the clearance function of human macrophages

To investigate whether our findings translate to human macrophages, we cultured primary human BMDM and treated them with either CM from primary human CD34$^+$ cells cultures (from healthy donors) or LCM obtained following culture of CD34$^+$ CML patient samples. Similar to what we observed in our mouse BMDM model, LCM significantly reduced efferocytosis compared to macrophages exposed to CM (Fig. 7a). Interestingly, supplementing normal CM with LTF or SSO resulted in a significant reduction in efferocytosis by human BMDM (Fig. 7a).

Complimentary, due to limited availability of human BM samples, we employed an established model of human macrophages by utilising THP-1 cell line-derived macrophages. Reassuringly, LTF treatment significantly reduced both efferocytosis and phagocytosis in THP-1-derived macrophages (Fig. 7b, c). Additionally, exposure to SSO resulted in a significant decrease in clearance of apoptotic cells without affecting phagocytosis of latex beads. Of note, no additive effect was observed when LTF was combined with the SSO inhibitor (Fig. 7b, c). Next, we investigated the effect of LTF and CML cell-derived LCM on CD36 expression. As we observed in the mouse model, we identified that both human LCM and LTF treatment significantly reduced the cell surface expression of CD36 on human macrophages (Fig. 7d, e). Additionally, LTF exposure decreased phosphorylation of the p65 subunit of NF-κB, p38 MAPK and ERK1/2 in control THP-1 macrophages in both the presence or absence of LPS (Supplementary Fig. 15a). Furthermore, we also observed that following CRIPSR-Cas9-mediated CD36 knockout (KO), THP-1-derived macrophages had a marked reduction in their efferocytosis capacity, to the same level of control THP-1 macrophages treated with LTF (Fig. 7f; Supplementary Fig. 15b, c). As seen with SSO exposure, loss of CD36 did not alter uptake of latex beads in THP-1-derived macrophages (Supplementary Fig. 15d). Interestingly, LTF exposure and loss of CD36 resulted in decreased basal and ATP-linked respiration, whereas 24 h SSO exposure did not have any significant effect on mitochondrial respiration (Supplementary Fig. 15e, f).

Finally, as we identified in the mouse model of CML that macrophages exposed to the leukaemic environment displayed a more immature phenotype (Fig. 3d, e; Supplementary Fig. 6a), we sought to investigate the effect of LTF and CD36 suppression on THP-1 CD11b expression. Exposure to LTF, and genetic or pharmacological inhibition of CD36, significantly decreased surface CD11b expression, implying that elevated LTF secretion and CD36 suppression contributes to the immature phenotype of macrophages (Fig. 7g, h).

## Discussion

CML remains a significant clinical burden due to LSC therapy persistence and disease relapse[24,25]. Exploration into the leukaemia BM niche is pivotal to our understanding of the environment that supports LSCs and potential ways to target this in the future. Our model has confirmed recent work of others that induction of CML can alter non-transformed neighbouring cells. Previously it has been demonstrated that bystander BCR::ABL1$^-$ stem cells display LSC signatures both in mouse models[11] and in patient samples[12]. Welner and colleagues also previously demonstrated a myeloid bias in Ph$^-$ cells in the CML niche, which was attributed to an enhanced production of IL-6 by CML cells[11]. Here we further characterised the myeloid expansion in the non-transformed haematopoietic compartment, revealing an increase in monocytes, macrophages and neutrophils at the expense of CD8$^+$ T cells and CD19$^+$ B cells. In addition to supporting previous findings that the CML niche supports a myeloid bias in untransformed cells, we uncover that Ph$^-$ macrophages are not only increased in frequency in the CML BM niche but are significantly impacted functionally. Specifically, we observed that Ph$^-$ macrophages have a significantly reduced expression of the scavenger receptor CD36 upon exposure to the CML microenvironment, uncovered by applying both CML mouse model and primary human macrophages. We show that macrophages exposed to CML have a reduced efferocytosis and phagocytic capacity upon exposure to CML-secreted factors, suppression of inflammatory gene transcript levels, a reduction in mitochondrial respiration and display a more immature phenotype.

In the field of AML, the LSC-macrophage phagocytosis axis has been demonstrated to be critical to the engraftment ability in xenograft models[20] and enhanced expression of CD47 renders AML cells resistant to phagocytosis[26]. Our findings appear to support the notion that in the leukaemia niche, the leukaemia immune surveillance and clearance by macrophages is suppressed. Here we show that this reduced clearance function appears to be mediated by suppressed CD36 expression, which may in part be driven by enhanced LTF production from CML stem/progenitor cells. Apart from its role in recognition and uptake of apoptotic cells, CD36 is a scavenger receptor that has a well-recognised role in metabolism. CD36 regulates lipid uptake, fatty acid oxidation (FAO) and mitochondrial metabolism[27–29]. It has been described that optimal alternative activation of macrophages depends on CD36, with loss of CD36 resulting in decreased FAO and oxygen consumption[27]. Similarly, we demonstrate that genetic inhibition of CD36, and to some extent exposure to LTF, suppresses mitochondrial respiration in THP-1-derived macrophages. However, it should be noted that the reduction is relatively modest and short-term SSO treatment did not alter mitochondrial metabolism in THP-1-derived macrophages.

The role of macrophages in cancer biology is complex, with significant heterogeneity in tumour-associated macrophages reported in multiple solid tumour types[30,31] and some haematological malignancies[32,33]. In models of T-ALL, leukaemia-associated macrophages have been shown to express a spectrum of genes related to both inflammatory and anti-inflammatory functions including CD206$^+$ macrophages expressing higher levels of IL-1β, IL-6 and Arg1, compared to CD206$^-$ counterparts[32]. This spectrum of genes has been

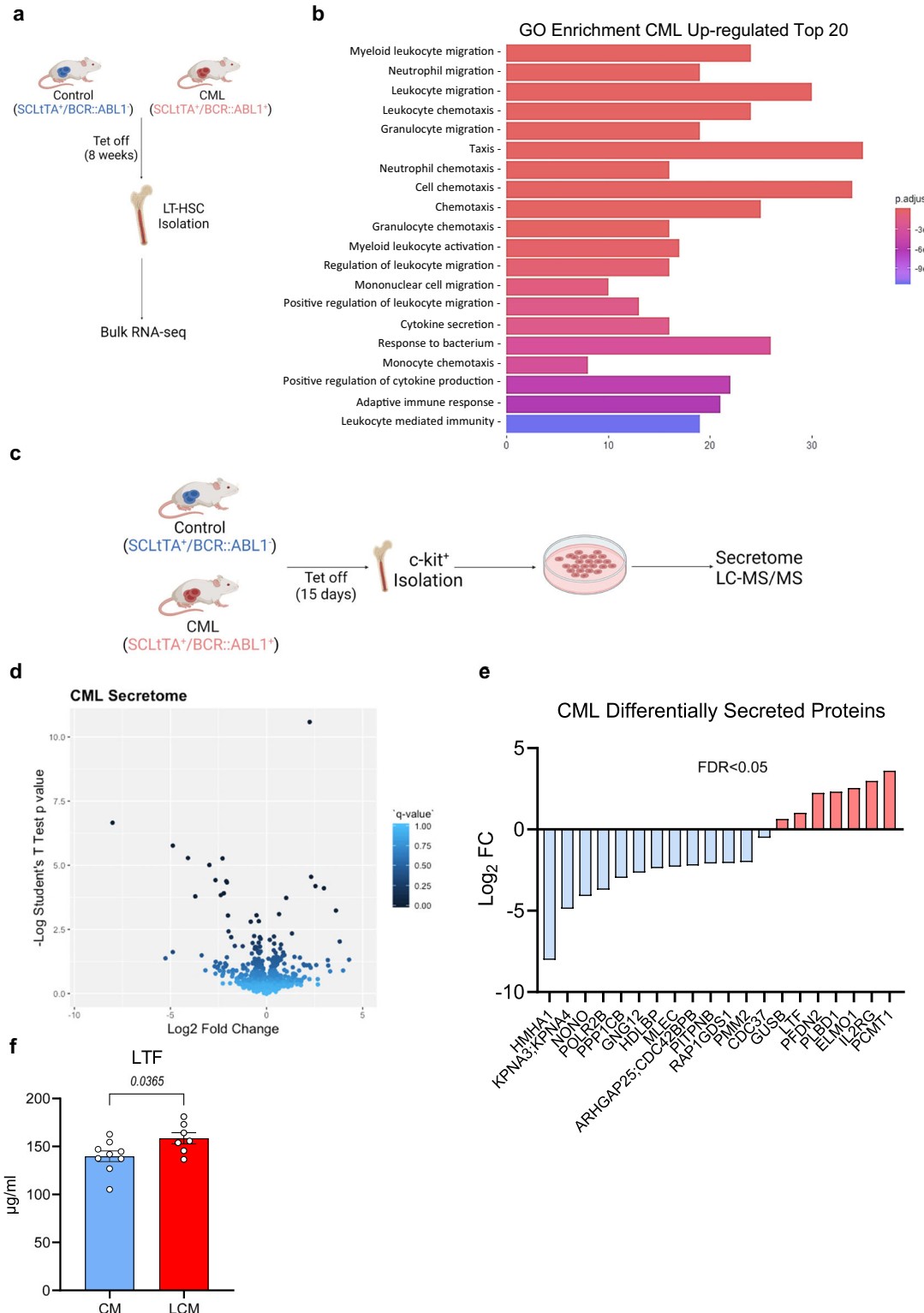

**Fig. 5 | CML stem/progenitor cells have a dysregulated protein secretome.**
**a**, **b** Schematic diagram of BM LT-HSC RNA sequencing experimental setup and gene ontogeny (GO) enrichment analysis of significant differentially expressed genes (*FDR < 0.05*) in CML LT-HSCs (*n* = 4 mice per experimental arm). Schematic created with BioRender.com (Agreement number: QN268HV791). **c** Schematic diagram of c-kit isolation and conditioned medium generation for secretory proteomics (MS). created with BioRender.com (Agreement number: HF268HUP38). **d** Volcano plot representing log$_2$ fold change between CML and WT mice against log$_2$ *p*-value for secreted proteins (*N* = 3 mice per experimental arm). **e** Log$_2$ fold change (CML/Control) of significantly changed proteins (FDR < 0.05) in CML vs WT. **f** Lactotransferrin (LTF) ELISA in c-KIT⁺ conditioned medium (48 h) from SCLtTA⁺/BCR-ABL⁻ (*n* = 9 conditioned medium samples) or SCLtTA⁺/BCR-ABL⁺ (*n* = 7 conditioned medium samples). Data are presented as the mean ± s.e.m. *P*-values were calculated using DESeq2 pairwise two-tailed *t*-test (*fold change < −1 or > 1; p < 0.05*) and the Benjamini−Hochberg method to correct for multiple comparisons (**b**), unpaired two-tailed *t*-test (**f**), and paired two-tailed *t*-test (**e**). Source data are provided as a Source Data file.

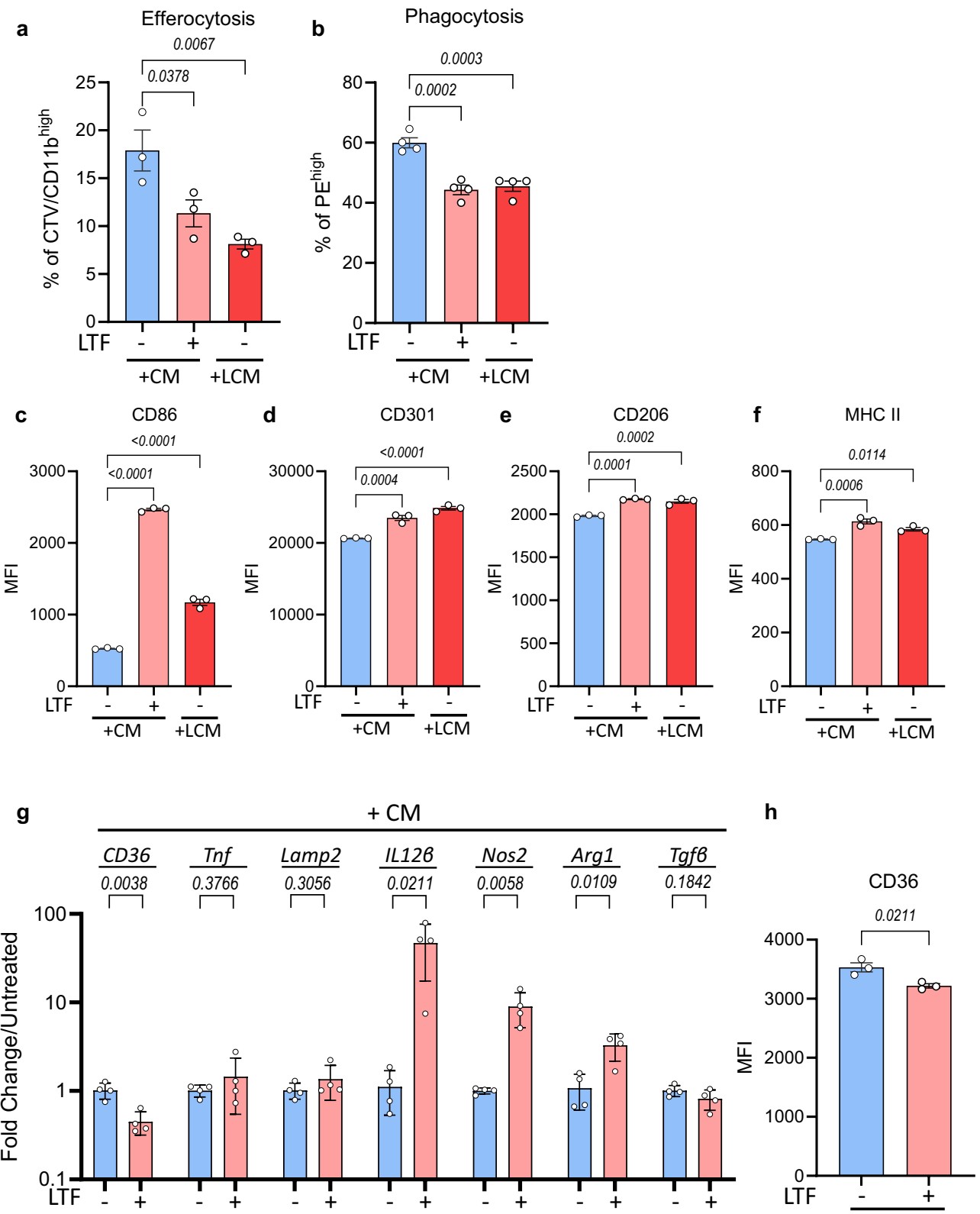

shown to be driven by organ-specific environments and has also been associated with solid tumour-associated macrophages[32]. However, the effects of the cancer microenvironment on oncogene-negative macrophages in CML remains largely unknown. In this study we have uncovered subpopulations of Ph⁻ macrophages in the CML niche that display distinct transcriptional profiles from control counterparts. scRNA-seq analysis has revealed that two unique subpopulations of Ph⁻

macrophages are present in the CML BM niche that display a range of heterogenic phenotypes with regards to maturity markers, phagocytosis, oxidative phosphorylation genes and inflammatory signatures. We have uncovered in both bulk and single-cell analysis that CML-exposed macrophages have low expression of *cd36*, and in single-cell clusters, low expression of *Nr4a1*, a factor essential to phagocytosis (Fig. 2h–j). These transcriptional signatures combined with our

**Fig. 6 | LTF exposure suppresses clearance function and CD36 expression in mouse macrophages. a** Quantification of uptake of apoptotic cells in BMDM conditioned with CM, CM in the presence of 50 µg/mL LTF, or LCM for 24 h (*n* = 3 independent experiments). **b** Quantification of phagocytosis of latex beads in BMDM following culture with CM in the presence or absence of 50 µg/mL LTF or LCM for 24 h (*n* = 4 independent experiments). **c–f** Surface marker expression of CD86 (**c**), CD301 (**d**), CD206 (**e**) and MHC II (**f**) in CM-exposed BMDM with or without 50 µg/mL LTF or LCM exposed BMDM following 24 h incubation (*n* = 3

independent experiments). **g** Relative log fold change in mRNA expression levels in BMDM exposed to CM with or without the addition of 50 µg/mL LTF for 24 h (*n* = 4 independent experiments). **h** Surface marker expression of CD36 in BMDM following exposure to CM in the presence or absence of 50 µg/mL LTF for 24 h (*n* = 3 independent experiments). Data are presented as the mean ± s.e.m. *P*-values were calculated using ordinary one-way ANOVA with Dunnet's multiple comparisons test (**a–f**), and unpaired two-tailed *t*-test (**g**, **h**). Source data are provided as a Source Data file.

findings that CML conditioned medium and co-culture experiments reduce macrophage phagocytosis of latex beads and efferocytosis of apoptotic cells, support that exposure to the CML niche reduces the clearance function of macrophages. Furthermore, we have identified *cd36* as a candidate marker that is downregulated in CML-exposed macrophages both at single-cell and bulk population level in CML (Fig. 2h–j) and genetic knockout of this receptor emulates reduced phagocytic function of macrophages. Future work should aim to investigate if CD36 expression can be used to monitor macrophage immune function and homoeostasis in CML progression or during TKI treatment.

The CML niche has been documented to be an inflammatory environment with high levels of IL-6[11], IL-1[34], and TNF[35], which complements our findings that cytokine signatures are enriched in CML LT-HSC RNA sequencing data (Fig. 4b). However, many proteins have immune modulatory functions outside classical cytokine classification that could be responsible for altering macrophage phenotype in the CML niche. Therefore, in our investigation we performed untargeted secretory proteomics to identify proteins differentially secreted by CML stem/progenitor cells compared to control. Here we uncovered that mouse CML c-Kit[+] cells secrete significantly more LTF, a known immune modulatory protein[36], when compared with secretion from normal counterparts (Fig. 4e, f). Interestingly, it has been reported that LTF impairs responsiveness to inflammatory stimuli and maturation of dendritic cells[37]. Furthermore, LTF also dampens immune response by blocking NF-κB nuclear translocation and NF-κB binding to the TNF promotor, subsequently suppressing pro-inflammatory gene transcription in THP-1 cells and human blood monocytes exposed to LPS[23]. Similarly, LTF exposure reduced NF-κB and MAPK signalling in BMDM and THP-1 macrophages. Furthermore, we observed that supplementing LTF to CM reproduces the reduction in CD36 expression, phagocytosis and efferocytosis, in manner similar to exposure to LCM. Additionally, LTF suppresses CD11b expression in THP-1-derived macrophages. Thus, we propose that LTF secretion from CML cells can at least in part be attributable to the altered function and immature phenotype of normal bystander macrophages in the CML niche. It should however be noted that LTF is likely one of many factors contributing to the observed cellular phenotype in this intricate environment of cytokines, chemokines, secreted immunomodulatory proteins and the complex array of cellular interactions. Furthermore, neutrophil granules have been found to contain LTF[38], thus we cannot exclude that mature cells in the bone marrow niche secrete LTF in a similar manner to primitive CML cells. Interestingly, LTF has been described to act as a tumour suppressor in several models of solid tumours including breast and nasopharyngeal carcinomas[39,40]. However, in our model LTF is likely to work as a tumour promoter by impairing the maturation and function of Ph⁻ macrophages. The autocrine effect of LTF on LSCs requires further investigation.

Finally, the study of macrophages in the human disease setting is exceptionally complex due to challenges in isolating macrophages that are not derived from the malignant clone. Here this study has shown that our findings in the elegant mouse model of CML are translatable to human in vitro models. It is worth noting that while THP-1 cells are a model of human macrophages in vitro, THP-1 cells are MLL-AF9[+]. Expression of the MLL-AF9[+] oncogene may be a contributing factor to the phenotype observed in this study and, as such, the results should

be interpreted with this in mind. However, in an attempt to address this shortcoming, we were able to demonstrate that macrophages derived from normal human bone marrow also displayed significantly reduced efferocytosis.

Overall, our findings have made significant steps in unravelling the complexities of the behaviour and phenotype of CML niche macrophages, a topic that may be previously unexplored both at a bulk population level or at single-cell resolution. Our results indicate that the CML microenvironment alters the capacity of macrophages to act as immune effectors in response to CML, partially through aberrant protein production by CML stem/progenitor cells. By identifying macrophage surface markers in the leukaemia environment and demonstrating reduced efferocytosis, phagocytic and metabolic function in the niche, we anticipate these findings to aid our understanding of the innate immune system in CML and have the potential to act as biomarkers for monitoring the effect of treatment in restoring immune homoeostasis.

## Methods
### Ethical approval
All patient samples were kindly donated with ethical approval and informed consent in agreement with the Declaration of Helsinki and with the approval of the National Health Service (NHS) Greater Glasgow and Clyde Institutional Review Board. Ethical approval was granted to the research tissue bank (REC 15/WS/0077) and for using surplus human tissue in research (REC 10/S0704/60).

All mouse experiments were performed in accordance with Home Office regulations and under approved project licences PPL PP2518370, PD6C67A47 and personal licences PIL IC9AB0748, I1F599357.

### Cells and cell culture
THP-1 and K562 cells were cultured in Roswell Park Memorial Institute (RPMI) (Cat# 11875093, Thermo Fisher Scientific) supplemented with 10% heat-inactivated foetal bovine serum (FBS) (Cat# 10100147, Thermo Fisher Scientific) 100 IU/mL penicillin/streptomycin (Cat# 15140122, Thermo Fisher Scientific), 2 mM L-glutamine (Cat# A2916801, Thermo Fisher Scientific) at 37 °C, 5% CO₂. THP-1 differentiation into macrophages was induced by addition of 20 ng/mL PMA (Cat# 1201, Biotechne) for 48 h. PMA was removed prior to any experimental procedures.

Primary CML samples were donated from individuals in chronic phase CML at the time of diagnosis and were product of leukapheresis. Equal number of male (2) and female (2) patient samples were used. The median age at diagnosis of patients was 58.5 years old, with ages ranging from 27 to 63 years old. Patient information is available in Supplementary Table 3. Normal samples were (i) surplus cells collected from femoral-head bone marrow, surgically removed from patients undergoing hip replacement or (ii) leukapheresis products from individuals with non-myeloid Philadelphia negative haematological disorders. CD34[+] cells were isolated using the CD34 MicroBead Kit or CliniMACS (Cat# 130-100-454, Miltenyi Biotec).

Primary CD34[+] cells were cultured in serum-free medium (SFM) consisting of Iscove's Modified Dulbecco's Medium (IMDM) (Cat# 12440061, Thermo Fisher Scientific) supplemented with 20% bovine serum albumin (BSA), insulin, and transferrin (BIT) (. Cat# 09500,

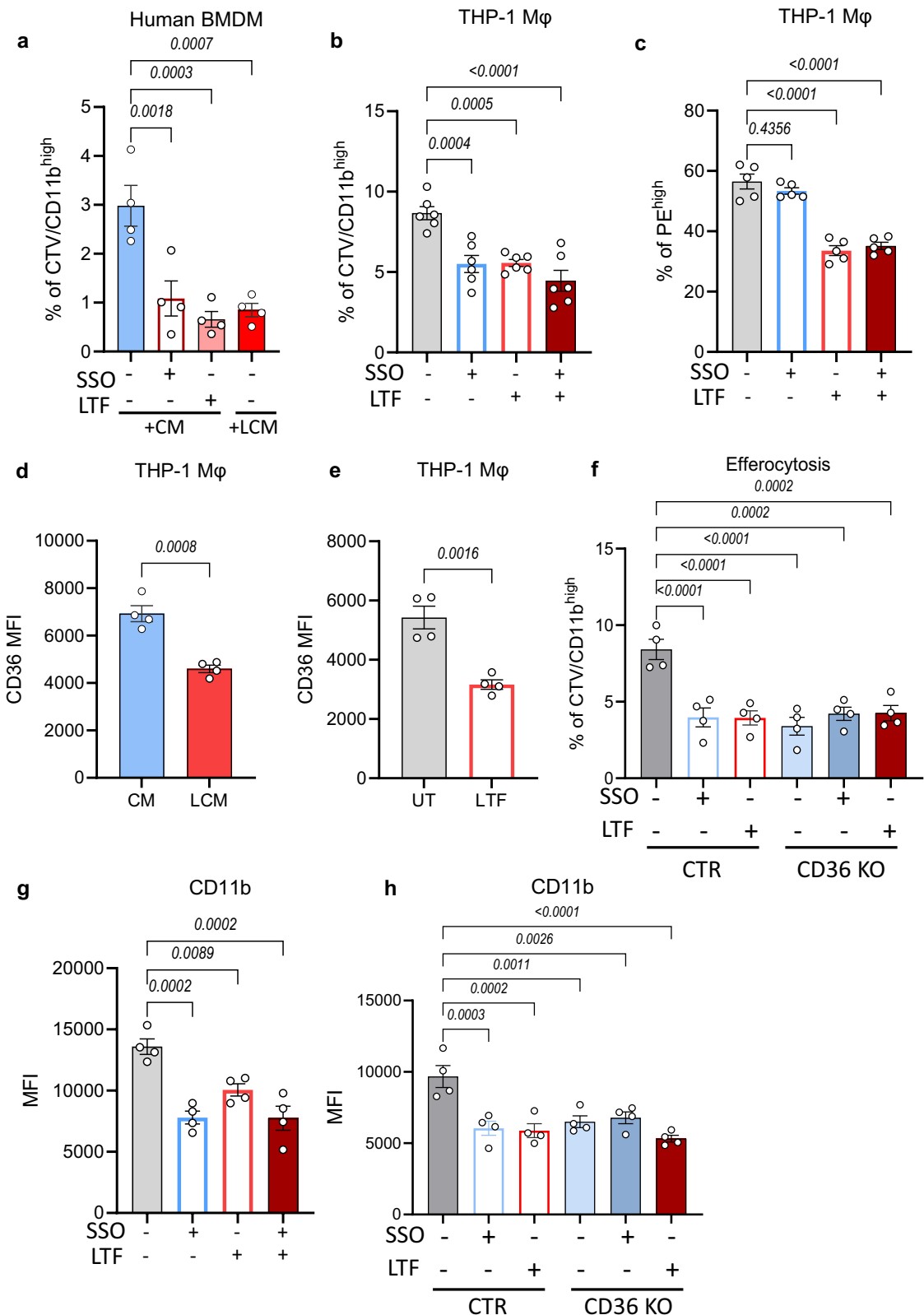

STEMCELL Technologies), low-density lipoprotein (40 µg/mL) (Cat# L4646, Sigma-Aldrich), 2-mercaptoethanol (0.1 mM) (Cat# 31350-010, Thermo Fisher Scientific) and 100 IU/mL penicillin-streptomycin. Cells were also supplemented with a physiological growth factor cocktail consisting of 0.2 ng/mL stem cell factor (SCF) (Cat# 573902, BioLegend), 0.2 ng/mL granulocyte macrophage colony-stimulating factor (GM-CSF) (Cat# 578602, BioLegend), 1 ng/mL interleukin-6 (IL-6) (Cat#

570802, BioLegend), 0.2 ng/mL macrophage inflammatory protein (Cat# 300-08, MIPα) (PeproTech) and 0.05 ng/mL leukaemia inhibitory factor (LIF) (Cat# 300-05, PeproTech).

Mouse BM was isolated from the hips and hind limbs of wild-type (WT) C57/Bl6 mice through manual crushing in 2%FBS/PBS. BMDM were generated with addition of mouse M-CSF (20 ng/mL) (Cat# 315-02, Peprotech) in Iscove's Modified Dulbecco's Medium (IMDM)

**Fig. 7 | LTF exposure alters the clearance function of human macrophages.**
**a** Quantification of clearance of apoptotic cells in human BMDM exposed to CM, CM in the presence of 50 μM SSO or 50 μg/mL LTF, or LCM for 24 h ($n = 4$ independent experiments). **b** Percentage of CTV/CD11b$^{high}$ THP-1-derived macrophages following a 24 h exposure to 50 μM SSO, 50 μg/mL LTF, or a combination of both ($n = 6$ independent experiments). **c** Quantification of phagocytosis of latex beads in THP-1 macrophages treated with 50 μM SSO, 50 μg/mL LTF, or a combination of both ($n = 5$ independent experiments). **d** Surface marker expression of CD36 in CM or LCM exposed THP-1-derived macrophages for 24 h ($n = 4$ independent experiments). **e** Expression of CD36 in untreated (UT) THP-1 macrophages or exposed to 50 μg/mL LTF ($n = 4$ independent experiments). **f** Percentage of CTV/CD11b$^{high}$ THP-1 control (CTR) or CD36 KO macrophages treated with 50 μM SSO or 50 μg/mL LTF for 24 h ($n = 4$ independent experiments). **g** Expression of CD11b in THP-1-derived macrophages treated with 50 μM SSO or 50 μg/mL LTF, or a combination of both for 24 h ($n = 4$ independent experiments). **h** Quantification of CD11b MFI in THP-1 CTR or CD36 KO macrophages incubated with 50 μM SSO or 50 μg/mL LTF for 24 h ($n = 4$ independent experiments). Data are shown as the mean ± s.e.m. *P*-values were calculated using ordinary one-way ANOVA with Dunnet's multiple comparisons test (**a–c**, **f–h**), and unpaired two-tailed *t*-test (**d**, **e**). Source data are provided as a Source Data file.

containing 20% FBS, 100 IU/mL penicillin/streptomycin for 7 days culture at 37 °C, 5% $CO_2$.

Human BM was isolated from the femur head of individuals undergoing elective hip replacement surgery through manual crushing. To isolate white blood cells, the BM was separated in a histopaque density gradient. To generate BMDM, the mononuclear cell layer was then suspended in Dulbecco's modified Eagle medium (DMEM) containing 10% FBS, 100 IU/mL penicillin/streptomycin and 20 ng/mL recombinant human M-CSF (Cat# 300-25, Peprotech) for 7 days culture at 37 °C, 5% $CO_2$.

### CD117 (c-Kit$^+$) isolation
Tetracycline was removed from SCLtTA$^+$/BCR::ABL1$^+$ (CML) and SCLtTA$^+$/BCR::ABL1$^-$ (Control) mice for 15 days. CD117$^+$ cells were isolated from BM using CD117 microbead kit (Cat# 130-097-146, Miltenyi Biotech). CD117$^+$ cells were cultured at 37 °C, 5% $CO_2$ in SFM consisting of IMDM supplemented with 20% BIT 9500, mouse IL-6 (25 ng/mL) (Cat# 575702, Biolegend), mouse IL-3 (10 ng/mL) (Cat# 575502, Biolegend), mouse SCF (50 ng/mL) (Cat# 579702, Biolegend), 0.2% 2-Mercarptoethanol, low-density lipoproteins (40 μg/mL).

### Mouse models
Animals of both sexes were housed in a pathogen-free facility at the Cancer Research UK Scotland Institute and kept in day/night cycles (12 h each), with temperature of 20–24 °C and humidity 45–65%. Mice were fed with ad libitum with food and water.

Single-cell macrophage experiment: Inducible chimeric SCLtTA$^+$/BCR::ABL1$^+$ BM was generated by mixing $7.5 \times 10^5$ BM cells from SCLtTA$^+$/BCR::ABL1$^+$ (CD45.2) CML, or SCLtTA$^+$/BCR::ABL1$^-$ (CD45.2) control, with $7.5 \times 10^5$ BM cells from C57/Bl6 mice (CD45.1) WT. C57/Bl6 recipient mice aged between 8 and 12 weeks received two doses of 4.25 Gy, 3 h apart and received $1.5 \times 10^6$ chimeric BM cells via tail vein injection. To allow for irradiation recovery, mice were maintained on tetracycline (0.5 g/L) for 14 days prior to removal of tetracycline for transgene induction. Disease levels were monitored for Gr-1 (Ly6G/Ly6C)-APC (Clone RB6-8C5, Cat# 108412, RRID: AB_313377, Biolegend, 0.5 μL/test) and CD11b-PE (clone M1/70, Cat# 101208, RRID: B_312791, Biolegend, 0.5 μL/test) in blood by flow cytometry. Mice were sacrificed at day 15 following BCR::ABL1 induction.

Bulk macrophage experiment: Inducible chimeric SCLtTA$^+$/BCR::ABL1$^+$ BM was generated by mixing $7.5 \times 10^5$ BM cells from SCLtTA$^+$/BCR::ABL1$^+$ (CD45.2) with $7.5 \times 10^5$ BM cells from C57/Bl6 mice (CD45.1) WT. C57/Bl6 recipient mice aged between 8 and 12 weeks received two doses of 4.25 Gy, 3 h apart and received $1.5 \times 10^6$ chimeric BM cells via tail vein injection. To allow for irradiation recovery, mice were maintained on tetracycline (0.5 g/L) for 14 days prior to removal of tetracycline for transgene induction. Disease levels were monitored for Gr-1 (Ly6G/Ly6C) and CD11b in blood by flow cytometry. Mice were sacrificed at day 15 following BCR::ABL1 induction.

Macrophage depletion using Anti-CSF1R: C57/Bl6 recipient mice transplanted with SCLtTA$^+$/BCR::ABL1$^+$ BM and WT BM (as described above) were treated with 400 μg/mouse anti-CSF1R (clone AFS98, Cat# BE0213, Bio-X-Cell) via the intraperitoneal route one day prior to

transgene induction. Following BCR::ABL1 induction anti-CSF1R was administered three times a week. Isotype control IgG2a (clone 2A3, Cat# BE0089, Bio-X-Cell) was used at 400 μg/mouse. Mice were euthanized according to institutional guidelines before reaching the threshold maximum weight loss.

### SSO and LTF treatment
Mouse/human BMDM and THP-1-derived macrophages were cultured with 50 μM SSO (Cat# 16464718, Fisher Scientific), 50 μg/mL LTF from bovine milk (Cat# L9507, Sigma-Aldrich), or a combination of both for 24 h.

### Flow cytometry
Cell suspensions were prepared from the BM isolated from the hind limbs of chimeric mice in 2% FBS/PBS solution. Cells were stained (100 μL/test) with monoclonal antibodies to mouse CD45.1-FITC (clone A20, Cat# 110705, RRID: AB_313495, Biolegend, 0.5 μL), CD45.1-PE (clone A20, Cat# 110707, RRID: AB_313496, Biolegend, 0.5 μL) CD45.2-PB (clone 104, Cat# 109820, RRID: AB_492873, Biolegend, 0.5 μL), CD45.2-PerCp/Cy5.5 (clone 104, Cat# 109827, RRID: AB_89335, Biolegend, 0.5 μL), CD11b-PE (clone M1/70, Cat# 101208, RRID: B_312791, Biolegend, 0.5 μL), Biotin-CD11b (clone M1/70, Cat# 553309, RRID: AB_394773, BD Biosciences, 0.5 μL), F4/80-APC/Cy7 (clone BM8, Cat# 123117, RRID: AB_893489, Biolegend, 0.5 μL), Ly6G-APC (clone 1A8, Cat# 127613, RRID: AB_1877163, Biolegend, 0.5 μL), Ly6G-PB (clone 1A8, Cat# 127612, RRID: AB_2251161, Biolegend, 0.5 μL), CD3-APC (clone 17A2, Cat# 100235, RRID: AB_2561456, Biolegend, 0.5 μL), CD4-PE (clone GK1.5, Cat# 100408, RRID: AB_312693, 0.5 μL), CD8a-PE (Clone 53-6.7, Cat# 100707, RRID: AB_312746, 0.5 μL), CD19-APC/Cy7 (clone 6D5, Cat# 115530, RRID: AB_830707, Biolegend, 0.5 μL), Lineage cocktail-PB (clone 17A2; RB6-8C5; RA3-6B2; Ter-119; M1/70, Cat# 133310, RRID: AB_11150779, Biolegend, 10 μL), SCA-1-Pe/Cy7 (clone D7, Cat# 108114, RRID: AB_493596, 1 μL) CD117 (c-Kit)-APC/Cy7 (clone 2B8, Cat# 105812, RRID: AB_313221, Biolegend, 1 μL), CD48-PE (clone HM48-1, Cat# 103406, RRID: AB_313021, Biolegend, 1 μL), CD150-APC (clone TC15-12F12.2, Cat# 115910, RRID: AB_493460, Biolegend, 1 μL) and CD11c-PE/Cy7 (clone N418, Cat# 117318, RRID: AB_493568, Biolegend, 1 μL). Following exposure to indicated treatments, mouse BMDM were lifted and stained (100 μL/test) with monoclonal antibodies to mouse CD11b-PE (clone M1/70, Cat# 101208, RRID: B_312791, Biolegend, 0.5 μL), CD36-PE/Cy7 (clone HM36, Cat# 102616, RRID: AB_2566122, Biolegend, 0.5 μL), CD86-APC (clone GL-1, Cat# 105011, RRID: AB_493342, Biolegend, 0.5 μL), CD206-AlexaFluor488 (clone C068C2, Cat# 141710, RRID: AB_10900445, Biolegend, 0.5 μL), CD301-PE/Cy7 (clone LOM-14, Cat# 145705, RRID: AB_2562940, Biolegend, 0.5 μL) and MHC II-PerCP/Cy5.5 (clone AF6-120.1, Cat# 116415, RRID: AB_1953308, Biolegend, 0.5 μL). Similarly, human BMDM or THP-1 macrophages were stained with human CD11b-PE (clone M1/70, Cat# 101208, RRID: B_312791, Biolegend, 0.5 μL) and/or CD36-PE/Cy7 (clone 5-271, Cat# 336222, RRID: AB_2716142, Biolegend, 0.5 μL). Stained cells were analysed on FACSVerse Flow Cytometer (BD Biosciences). Analysis was conducted using FloJo (v.10.7). Data represented as percentage of whole BM and absolute cell number calculated from whole BM counts.

## Conditioned medium generation

CD117 enriched BM samples, primary CD34$^+$ samples and, CML and AML cell lines were seeded at $0.2 \times 10^6$/mL in respective culture medium (as described above) and cultured for 48 h. Conditioned medium was harvested by two centrifugation steps at $300 \times g$ for 10 min, followed by $2000 \times g$ for 10 min at 4 °C. Supernatant was collected and stored at −80 °C until use.

## Conditioned medium generation for secretomic analysis

CD117 enriched BM samples were washed four times in sterile PBS and $1.2 \times 10^6$ cells per sample plated in 1 mL of serum-free IMDM containing 1% penicillin/streptomycin, mouse IL-3 (10 ng/mL) and mouse SCF (50 ng/mL). Following overnight incubation, 3 centrifugation steps at 4 °C, 10 min at $300 \times g$, 10 min at $2000 \times g$ and 30 min at $10,000 \times g$ were performed and the supernatants collected. Supernatant was acidified using 10% trifluoracetic acid to pH5 and incubated with Strataclean Resin (Agilent Technologies) for 1 h as previously described[41]. Proteins were eluted from the resin and samples were boiled for 5 min at 95 °C and run on a 4–12% gradient NuPAGE Novex Bis-Tris protein gel. The gel was washed in HPLC water, and then incubated in InstantBlue solution (Sigma) for 1 h. The gel was washed further in HPLC water and in-gel digested.

## Mass spectrometry analysis

Each gel lane was excised and reduced using 10 mM dithiothreitol, alkylated with 55 mM iodoacetamide, and digested with trypsin (Trypsin gold), overnight at 35 °C. Tryptic peptides were dried in a centrifugal evaporator. Tryptic peptides were analysed on Q-Exactive HF (Thermo Scientific) coupled online to an EASY-nLC II (Thermo Scientific). Dried tryptic peptides were re-suspended in and loaded with buffer A (2% acetonitrile, 0.1% formic acid). Peptides were then separated on a 20 cm fused silica emitter (New Objective) packed in-house with ReproSil-Pur C18-AQ 1.9 µm resin (Dr Maisch GmbH) using a flow of 200 nL/min in a 42 min gradient from 5% to 28% buffer B (80% acetonitrile, 0.1% formic acid), followed by 13 min gradient from 28% to 45% buffer B. The emitter was kept at 36 °C with a column oven integrated into the nanoelectrospray ion source (Sonation). The MS spectra were acquired in the Orbitrap analyser at a resolution of 60,000 at 400 m/z.

The MS files were analysed using MaxQuant software[42] (v. 1.5.51) and searched against the mouse UniProt database for protein identification. A false discovery rate (FDR) of 1% was used for peptide and protein identification. The label-free quantification (LFQ) algorithm in MaxQuant was used for protein quantification[43]. Those proteins identified with a minimum of one unique peptide and with at least two quantification events were used for the analysis. The data output was further filtered using Perseus software. Proteins used for downstream analysis were those quantified in three out of three biological replicates. To identify significantly changed proteins a two-sample Student's $t$-test was performed, with an FDR of 0.05.

## LTF ELISA

CD117 conditioned medium was analysed for LTF concentration using a mouse LTF ELISA kit (Cat# orb409330, Biorbyt) according to the manufacturer's instructions.

## Uptake of c-kit cells by BMDM

CD117$^+$ selected cells were stained with CellTrace Violet (CTV) (Cat# C34557, Thermo Fisher Scientific) according to the manufacturer's instructions. CD117$^+$ cells were washed 3× in PBS and cultured in c-kit complete medium with mouse BMDM for 48 h. C-Kit$^+$ cells were removed from the co-culture and macrophages washed 3× in PBS. BMDM were scraped from the plate and data were acquired on FACSVerse Flow Cytometer for CTV signal. Analysis was conducted using FloJo (v.10.7).

## Efferocytosis assay

Mouse or human BMDM and THP-1-derived macrophages were cultured with conditioned medium in the presence or absence of SSO or LTF for 24 h. K562 cells were stained with CTV and apoptosis was induced by a 45 s heat shock at 95 °C. Macrophages were stained with CD11b. Next, apoptotic cells were added to macrophages in a 3:1 ratio and incubated for 2 h. Medium and floating apoptotic cells were removed, and macrophages washed 2× in PBS. Macrophages were scraped from the plate and efferocytosis was assessed on FACSVerse Flow Cytometer. Data were analysed using FloJo (v.10.7).

## Phagocytosis assay

BMDM and THP-1-derived macrophages were cultured with conditioned medium in the presence or absence of SSO or LTF for 24 h. Fluorescent 1 µm microspheres (Cat# F13082, Thermo Fisher Scientific) were added to the cell culture for 4 h. Excess beads were removed by PBS wash and BMDM bead uptake was assessed and analysed as described above.

## RT-qPCR

BMDM were plated at $0.5 \times 10^6$ cells per well and cultured in CM or LCM for 24 h. RNA was extracted using PicoPure RNA Isolation Kit (Thermo Fisher Scientific). Reverse transcription was performed with the High-Capacity cDNA Reverse Transcription Kit (Thermo Fisher Scientific). The following primers were used against mouse beta-actin (Thermo Fisher Scientific: Mm02619580_g1) and mouse TNF (Thermo Fisher Scientific: Mm00443258_m1) with TaqMan Advanced Fast Master Mix (Thermo Fisher Scientific). The PCR was performed with the following steps: 2 min at 50 °C, 10 s at 95 °C, followed by 40 cycles at 95 °C for 15 s and 60 °C for 30 s on C1000 Touch Thermal Cycler (BioRad). Data were acquired with the CFX Maestro Software (v.1.1) (BioRad). The relative mRNA expression was calculated via the ΔΔCT method. For BMDM treated with LTF, RNA extraction and reverse transcription was performed as described above. The following mouse primers were used: CD36 Forward 5′ AGATGACGTGGCAAAGAACAG 3′ Reverse 5′ CCTTGGCTAGATAACGAACTCTG 3′; TNF Forward 5′ CAGGCGGTGCCTATGTCTC 3′ Reverse 5′ CGATCACCCCGAAGTTCA GTAG 3′; LAMP2 Forward 5′ TGGCTCAGCTTTCAACATTTCC 3′ Reverse 5′ TGCCAATTAGGTAAGCAATCACT 3′; IL-12β Forward 5′ TGGT TTGCCATCGTTTTGCTG 3′ Reverse 5′ ACAGGTGAGGTTCACTGTTTCT 3′; NOS2 Forward 5′ ACATCGACCCGTCCACAGTAT 3′ Reverse 5′ CAGAGGGGTAGGCTTGTCTC 3′; ARG1 Forward 5′ CTCCAAGC-CAAAGTCCTTAGAG 3′ Reverse 5′ AGGAGCTGTCATTAGGGACATC 3′; TGFβ Forward 5′ CTTCAATACGTCAGACATTCGGG 3′ Reverse 5′ GTAACGCCAGGAATTGTTGCTA 3′ and 18S Forward 5′ GTAACCCG TTGAACCCCATT 3′ Reverse 5′ CCATCCAATCGGTAGCG 3′. The cDNA products were mixed with Lightcycler 480 SYBR Green Master Mix (Roche) and the primers mentioned above. The PCR was performed with the following steps: 10 min at 95 °C, followed by 40 cycles of 15 s at 95 °C, 1 min 60 °C and 45 s at 72 °C. The relative mRNA expression was calculated via the ΔΔCT method with 18S used as internal control.

## Macrophage gene expression analysis by Fluidigm multiplex RT-qPCR

Bulk RNA from CD45.1$^+$CD11b$^+$F4/80$^+$ macrophages isolated from chimeric SCLtTA$^+$/BCR::ABL1$^+$ on/off tetracycline was reverse transcribed as described above. Specific target amplification of cDNA was performed with primers (see Supplementary Table 4) and multiplex pre-amp master mix (Qiagen) for 14 cycles. Amplified cDNA was cleaned with Exonuclease I treatment (NEB).

A 48.48 integrated fluidic circuit (IFC) (Fluidigm) was primed according to the manufacturer's instructions. Forward and reverse primers were prepared with EvaGreen Master Mix and loaded onto IFC. Pre-amplified cDNA samples were added to their respective IFC islets and loaded onto the BioMark HD. The RT-qPCR was performed with

the following steps: 95 °C for 60 s, 96 °C for 5 s followed by 20 s at 60 °C for 30 cycles, then a melt curve was performed at 60 °C for 3 s and ramp 60–95 °C at rate of 1 °C/3 s.

Data normalised relative to an average of β2 microglobulin (B2M) and 18 s expression. Fold change calculated relative to one WT control. Heatmap was generated on average expression of $n = 4$–7 mice.

## Macrophage gene expression analysis by single-cell RNA sequencing

BM cell suspensions were treated with Diisopropylfluorophosphate (1 mM) prior to cell sorting. Cell suspensions were stained with monoclonal antibodies to mouse CD45.1, CD45.2, CD11b, F4/80 and single-cell macrophages (CD45.1+ CD11b+ F4/80+) were sorted into 96-well plates containing 0.2% Triton X-100 and 0.1 µL of RNAase plus inhibitor (Promega N2615)/each well using BD FACS ARIA Z6001 (BD Biosciences).

Smart-seq2 Library preparation was performed as previously described[13]. In brief, oligo-dT30 primers and dNTP mix were mixed with isolated RNA and incubated with samples at 72 °C for 3 min. Reverse transcription and pre-amplification steps were performed. cDNA was then purified, and library quality assessed using Tapestation D5000. Samples were subsequently tagmented and amplified according to the Nextera XT DNA Library Preparation protocol. Indexed samples were then pooled and cleaned. Libraries were then sequenced at 2 × 75 bp paired end reads, with a total of 100 M reads.

## Single-cell bioinformatic data analysis

Sequencing was performed on the NextSeq500 (Illumina) by Polyomics at the University of Glasgow. The sequences were checked for quality using FastQC v0.11.8[44] and cells with mean quality under 30 in most sequences were eliminated. The sequences were trimmed with Trim_galore v0.5.0[45], using the default threshold for quality and aligned to the reference genome (GRCm38, obtained via Ensembl on 10/12/2017 with ERCC sequence – Thermo-Fischer, #4456739 – manually added) using Hisat2 v2.1.0[46]. Cells with an alignment score lower than 50% were discarded. A count matrix was produced with FeatureCounts v1.6.2[47].

Data analysis was performed using Scater[48] and Scran[49] packages in R[50]. Quality control measures were performed to remove cells that fell below the quality threshold for total counts, the number of features detected, those with a high percentage of mitochondrial genes and ERCC. A median absolute deviation of 4 was used as a threshold for quality control. This resulted in an ERCC threshold of 10.539%, mitochondrial content threshold of 8.433%, sum of counts threshold of 5137.281 and feature number threshold of 96.785. Counts were normalised by calculating their $\log_{10}$. Cell clustering was also mapped to plates processed (Plates A–D). This clustering analysis revealed that there was no plate-specific effect driving the observed cell clustering. Clustering was performed using nearest neighbour from the Scran package followed by cluster walktrap from the igraph package[51]. T-SNE plots were produced using Scater. Markers for clusters were obtained using the findMarkers function in Scran and heatmaps were plotted using Scater. Violin plots were generated using ggplot2[52].

Differential expression was calculated using DeSeq2[53] and Zinbwave[54], with each cluster being compared to the average of all the other clusters together (example: cluster1/(0.2 * cluster2 + 0.2 * cluster3 + 0.2 * cluster4 + 0.2 * cluster5 + 0.2 * cluster6)). GSEA[55,56] was performed on a ranked list formed of the -log of the p-value of each gene, multiplied by −1 in case the log2 fold change was negative, or 1, in case the log2 fold change was positive. Pathway analysis using Kegg[57–59], Reactome[60,61] and Wikipathways[62] was performed using http://www.webgestalt.org (accessed in December 2020)[63–65], and bar plots were generated using ggplot2.

For the human scRNA-seq dataset, data analysis was performed using Seurat[66] package in R (v.4.2.2). Quality-controlled datasets of 163,146 single cells were obtained from Krishnan et al.[20] for further downstream analyses. Counts were log-normalised and scaled. PCA was run to identify the dimensionality of the dataset. Using the top 20 principal components, nearest neighbours were identified; followed by clustering with a resolution of 0.25. UMAP plot was produced using 25 nearest neighbours, cosine metric, top 30 dimensions and default seed of 42.

Macrophage markers were plotted using FeaturePlot function in Seurat, to identify the myeloid cluster. The myeloid cluster was sub-clustered using the same pipeline with top 30 principal components and a resolution of 0.2. Further, macrophage markers in these sub-clusters were plotted to identify CSF1R+ macrophage cluster. This macrophage subpopulation was further clustered using the same pipeline with top 30 principal components and a resolution of 0.8. Finally, markers for clusters were obtained using the FindMarkers function in Seurat and volcano plots were generated using ggplot2.

## LT-HSC RNA sequencing analysis

LSK Flt3− CD150+ CD48− (LT-HSC) cells were sorted from BM of SCLtTA/BCR::ABL1 mice following 8 weeks doxycycline removal and from healthy, age- and sex-matched WT mice. RNA was extracted using RNeasy Plus Micro Kit (QIAGEN), with 4 biological replicates from each group. Sequencing libraries were prepared with the SMARTer Ultra Low Input RNA Kit for Sequencing (v4, TaKaRa Clontech) and the Nextera XT DNA Library Preparation Kit (96 samples, Illumina). Sequencing was performed using the HiSeq 2500 platform with the HiSeq SBS Kit V4 (Illumina). The quality of fastQ files was assessed using FastQC2 and sequences with a quality score of less than 20 were trimmed using Trim Galore![67] v0.4.4_dev. Adaptors were also trimmed using Trim Galore with stringency set to 5. Quality control was repeated on trimmed data using FastQC2. Sequences were aligned using Hisat2[68] to the reference genome (Ensembl release 92 primary assembly reference genome downloaded from Ensembl 5). The chromosome reference annotation (Ensembl) was also provided using the –known-splicesites parameter[68] to facilitate the alignment. Normalisation and differential expression were calculated using DESeq[53]. For gene ontogeny (GO) analysis the clusterProfiler package with the enrichGO function[69] was used to determine significantly enriched GO terms (FDR < 0.05) on genes considered significantly up (FC > 1, $p < 0.05$) or down (FC < −1, $p < 0.05$) regulated. Statistical analysis was performed using R Studio (v.1.1.4).

## Generation of stable knockout (KO) cell lines

To generate CRISPR–Cas9-mediated THP-1 CD36 KO cells, we used the following guide sequence: ACCTTTATATGTGTCGATTA. Synthesis of selected guide was conducted by Integrated DNA Technologies. Oligonucleotides were annealed and cloned in BsmBI–digested lenti-CRISPRv.2-puro plasmid (Addgene). Lentiviral particles for cloned lentiCRISPRv.2 were generated by transfecting human embryonic kidney (HEK) 293FT cells with psPAX2 packaging and pCMV-VSV-G envelope vectors using the calcium phosphate method[70]. Viral supernatant was harvested after 24 and 48 h and transferred onto THP-1 cells. Cells were then selected using 3 µM/mL puromycin and presence of stable KO was validated by immunoblotting.

## Immunoblotting

Cells were lysed using RIPA buffer (Cat# 89900, Thermo Fisher Scientific) containing cOmplete Protease (Roche), PhosSTOP phosphatase (Roche) inhibitors and 1% SDS. Protein concentration was quantified using a Pierce BCA protein assay kit (Thermo Fisher Scientific). Then, 10–20 µg of protein were resolved in 4–12% SDS–polyacrylamide gel electrophoresis gels and transferred to PVDF membranes. After blocking in 2% BSA in Tris Buffer Saline 0.01% Tween for 1 h, membranes were incubated overnight at 4 °C with primary antibodies: Phospho-NF-κB p65 (Ser536) (Clone 93H1, Cat# 3033, Cell

Signalling Technology, 1:1000), NF-κB p65 (Clone D14E12, Cat# 8242, Cell Signalling Technology, 1:1000), phospho-p38 MAPK (Thr180/Tyr182) (Clone D3F9, Cat# 4511, Cell Signalling Technology, 1:2000), p38 MAPK (Clone D13E1, Cat# 8690, Cell Signalling Technology, 1:2000), phospho-p44/42 MAPK (Erk1/2) (Thr202/Tyr204) (Clone D13.14.4E, Cat# 4370, Cell Signalling Technology, 1:2000), p44/42 MAPK (Erk1/2) (Clone 137F5, Cat# 4695, Cell Signalling Technology, 1:2000), CD36 (Clone EPR6573, Cat# ab133625, Abcam, 1:1000), H3 (Clone D1H2, Cat# 4499, Cell Signalling Technology, 1:2000) and GAPDH (Clone D16H11, Cat# 5174, Cell Signalling Technology, 1:2000). The following day, membranes were incubated with rabbit HRP-linked secondary antibody (Cat# 7074, Cell Signalling Technology, 1:3000) for 1 h and visualised using ECL reagent (Cat# RPN2235, Cytiva). Images were acquired with the LI-COR Odyssey FC imaging (LI-COR Biosciences) system with Image Studio software (v.5.2).

### Cellular energy metabolic analysis

**Metabolic analysis of BMDM.** BMDM were plated at 50,000 cells per well of a 96-well plate and allowed to adhere overnight. Wild-type and CML c-Kit$^+$ cells were obtained as described above in c-Kit isolation and 10,000 cells per well were plated directly on top of macrophages, culturing both in mouse SFM for 24 h. Following co-culture, c-Kit cells were removed by washing 3× in PBS. Minimal DMEM was prepared with glucose (25 mM) and glutamine (4 mM), pH adjusted to 7.4, and 175 µL added per well. A mitostress test was performed using XF96 flux analyser (Seahorse Bioscience). according to manufacturer's instructions. Injection drugs were prepared as 1.5 µM oligomycin, 1.5 µM CCCP, 1.25 µM rotenone and 2.5 µM antimycin A. Immediately after completion of the assay, protein extraction was performed, and measurements normalised to protein content of each well.

**Metabolic analysis of BMDM treated with LTF.** BMDM as prepared according to the method as described above were plated at 50,000 cells per well of a 96-well plate. Cells were plated in complete c-kit medium (untreated) or 50 µg/mL bovine LTF. Cells were cultured for 16 h. Minimal DMEM was prepared and mitostress test performed described above.

**Metabolic analysis of THP-1-derived macrophages.** THP-1-derived &¯ophages were plated at 50,000 cells per well of a 96-well in the presence of 20 ng/mL PMA for 48 h. Next, PMA removed, and cells were cultured in the presence or absence of SSO or LTF for 24 h. Minimal RPMI was prepared with the addition of glucose (11.1 mM) and glutamine (2 mM), pH adjusted to 7.4, and 175 µL added per well. A mitostress test was performed using XF instrument according to manufacturer's instructions. Injection drugs were prepared as 1 µM oligomycin, 1 µM CCCP, 1 µM rotenone and 1.0 µM antimycin A. Measurements were normalised using protein content.

### Statistical analysis

Statistical analysis was performed using GraphPad Prism 9 Software (v.9.3.1), unless otherwise specified. Statistical significance between conditions was calculated using an unpaired two-tailed *t*-test when comparing two groups, whereas one-way ANOVA was used to assess statistical significance between three or more groups with one experimental parameter. For multiple testing in Fluidigm gene expression datasets a multiple *t*-test with Benjamini and Hochberg multiple correction method was performed to account for multiple comparisons. For in vivo experiment with macrophage depletion, overall survival was monitored by Kaplan–Meier analysis, and *p*-values were calculated using log-rank (Mantel–Cox) test. Data are represented as mean ± standard error of mean (s.e.m.) as outlined in figure legends.

No statistical methods were used to predetermine sample size for in vitro experiments. Sample sizes were estimated according to common practice for each experimental design and preliminary and previous experiments to estimate variability. In vitro experiments were repeated at least on three separate occasions, except where noted. For experiments with primary samples, a minimum of four samples was used to give adequate power. For all in vitro experiments, unless stated otherwise, cells were randomly plated/treated/analysed during each experiment. Investigators were not blinded during data processing and analysis of in vitro data. For in vivo experiments, while no statistical methods were used to calculate the exact cohort sample size, the number of animals used per arm in each experiment was estimated based on variability of pilot and previous experiments. Similarly aged animals were transplanted with the same number of donor cells. All mice were cared for equally in an unbiased fashion by animal technicians and investigators, and no animal was excluded from the analysis. Animals were assigned to different groups in a manner ensuring a consistent average starting engraftment across the groups. For collection of in vivo data, investigators were not blinded to group allocation as the experimental design made this not feasible. For data analysis, investigators were blinded to the experimental conditions when assessing the outcomes, if feasible.

### Reporting summary

Further information on research design is available in the Nature Portfolio Reporting Summary linked to this article.

### Data availability

The expression profiling scRNA-seq data generated in this study have been deposited in public Gene Expression Omnibus (GEO) database under accession code GSE204978. Data request can also be made by contacting the corresponding authors (vignir.helgason@glasgow.ac.uk or kristina.kirschner@glasgow.ac.uk). Raw data used in Fig. 5a, b and Supplementary Figure 11 has been uploaded to GEO under accession code GSE126547[71] and is available to the scientific community/general public. Data request, including relevant codes, can also be made by contacting co-author Ravi Bhatia (rbhatia@uabmc.edu). Raw data used in Fig. 5d, e has been deposited to the ProteomeXchange Consortium via the PRIDE partner repository[72]. Dataset identifier is PXD022366. Raw data used in Supplementary Figure 9 are available online at the European Genome-Phenome Archive (EGA) under accession ID EGAS00001005509 upon written request to the data access committee. Using a R shiny APP, the scRNA-seq data can be interactively viewed at: http://scdbm.ddnetbio.com/. Data request, including relevant code can also be made by contacting co-author Ong Sing Tiong (sintiong.ong@duke-nus.edu.sg). The remaining data are available within the Article, Supplementary Information or from the corresponding author upon request. Source data are provided with this paper.

### Code availability

Relevant code has been deposited on GitHub and made publicly available: https://github.com/Kiron-J-Roy/Leukaemia-Exposure-Alters-the-Transcriptional-Profile-and-Function-of-BCR-ABL-Negative-Macrophages.

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

## Acknowledgements

The authors would like to thank the Core Services and Advanced Technologies at the Cancer Research UK Scotland Institute (C596/A31287), with particular thanks to the Biological Services Unit. We thank Dr Alison Michie for assisting with the in vivo research, Tom Gilbey for cell sorting and Lisa Neilson for assistance in proteomic sample preparation. We also thank Glasgow Polyomics for performing sequencing. Schematic graphs in Figs. 1a, 2a, 3a, 5a, c and Supplementary figure 4a were created with BioRender.com. This work was supported by Friends of the Paul O'Gorman Leukaemia Research Centre (A.D., G.V.H., J.S., Z.K.), The Howat Foundation (G.V.H.), The Kay Kendall Leukaemia Fund (KKL698, P.B., G.V.H.), Tenovus Scotland (A.I., G.V.H.), Cancer Research UK (A29800 to S.Z. and A29754 to G.V.H.).

## Author contributions

G.V.H. supervised the entire study. A.D. M.M.Z. and G.V.H. designed the study and wrote the manuscript. A.D. and M.M.Z. performed all in vitro experiments and supervised all in vivo work, analysed and interpreted data. B.P. and J.S. performed analysis of sequencing data. Y.C.H. assisted in sample preparation for sequencing. G.R.B. analysed and interpreted data for proteomics. D.Z., E.H., A.I., Z.K., P.A., P.B., K.D. assisted in sample preparation and in vivo work. I.V.L. and Z. K. assisted in in vitro experiments. M.S. and K.D. assisted with in vivo model experimental design. H.A., D.V. and M.C. provided material support. P.A. and R.B. provided bulk RNA sequencing data, L.M. and L.E.H. assisted in the analysis of this data. V.K. and O.S.T. provided human single-cell RNA sequencing dataset and assisted in the analysis. S.C. provided material support and contributed to the design and analysis of in vivo experiments. H.W. provided vital reagents and expertise for in vitro experiments. S.Z. provided essential reagents and supervised proteomic experiments. K.K. contributed to the design and supervised sequencing experiments.

## Competing interests

M.C. has received research funding from Cyclacel and Incyte, is/has been an advisory board member for Novartis, Incyte, Jazz Pharmaceuticals, Pfizer and Servier, and has received honoraria from Astellas, Novartis, Incyte, Pfizer and Jazz Pharmaceuticals. All other authors declare no competing interests.

## Additional information

[1]Wolfson Wohl Cancer Research Centre, School of Cancer Sciences, University of Glasgow, Glasgow G61 1QH, UK. [2]Paul O'Gorman Leukaemia Research Centre, School of Cancer Sciences, University of Glasgow, Glasgow G12 0ZD, UK. [3]Cancer Research UK Scotland Institute, Glasgow G61 1BD, UK. [4]Cancer & Stem Cell Biology Signature Research Programme, Duke-NUS Medical School, Singapore, Singapore. [5]Division of Hematology and Oncology, Department of Medicine, University of Alabama at Birmingham, Birmingham, AL, USA. [6]Department of Clinical Laboratory Sciences, College of Applied Medical Sciences, Najran University, Najran 61441, Kingdom of Saudi Arabia. [7]Universidad de Alcalá, Facultad de Medicina y Ciencias de la Salud, Dpto. de Biología de Sistemas, Unidad de Bioquímica y Biología Molecular, E-28805 Madrid, Spain. [8]These authors contributed equally: Amy Dawson, Martha M. Zarou. ✉e-mail: kristina.kirschner@glasgow.ac.uk; Vignir.Helgason@Glasgow.ac.uk

