## [Peer Review File · Nature Communications]

Leukaemia Exposure Alters the Transcriptional Profile and Function of BCR::ABL1 Negative Macrophages in the Bone Marrow NicheREVIEWER COMMENTS

Reviewer #1 (Remarks to the Author):

The manuscript presents experimental work using state-of-the-art methods studying the role of macrophages in CML on a single cell level. The authors used a chimeric CML mouse model to study the role CML cells on the marrow microenvironment and on local bystander macrophages. The major findings of the paper were:

- CML cells can alter macrophages transcriptomes and phagocytic function.
- CML exposed macrophages show down-regulation of phagocytosis genes CD36 and Lamp2.
- CML cells secrete factors including lactotransferrin, which suppress phagocytosis, mitochondrial respiration, and inflammatory genes.

Major concerns:

There are several major concerns with the manuscript:

1. All experiments were carried out with an (elegant) murine model or with murine cell lines. Therefore, the relevance of the findings for human disease remains unanswered. It would be highly desirable to conduct several experiments with human cells or tissues to create evidence for relevance in human CML.
2. Overall, the manuscript remains descriptive in large parts. For example, no mechanistic explanation is given on the regulatory molecules or signaling pathways that are induced in murine macrophages when exposed to CML cells or secreted factors (such as lactoferrin). No mechanistic explanation is given regarding the downregulation of some potentially relevant molecules such as CD36 or Lamp2.
3. Similarly, on a functional level, the manuscript could benefit from functional experiments confirming the findings by alternative approaches. For example, the claim (e.g., Figure 3a) that the reduced expression of CD36 is functionally relevant for the phagocytic function of macrophages in the bone marrow of CML could be tested by blocking of CD36 in CD36 positive macrophages or CD36 gene deletion experiments. These experiments could be carried out, e.g., with coculture of CML cells or in functional tests of phagocytosis.

Additional, major points:

Figure 2e: There is an apparent difference regarding the number of CD8+ T cells in the CML

compartment versus the CML exposed compartment. How do the authors explain this difference?

Figure 4a and b:

Can any of the clusters C1-C6 be found in human CML? It would be desirable to expose human macrophages to human CML cells (or supernatant) to investigate the relevance of the findings for human disease.

Figure 7a: could these experiments be reproduced with human macrophages or macrophage cell lines?

Figure 7g: In this figure, one can see significant changes only for TNFalpha and TGFbeta but not CD36, LAMP2 or others. This represents a contradiction to other experiments, e.g., of Figure 4d. This means that LTF only produces a subset of the changes seen with CML exposed cells. What other proteins of the secretome contribute to the observed effects?

Figure 7h: The oxygen consumption rate shown upon 24-hr LTF stimulation drops somewhat, but this seems a rather small change. Please comment whether these changes are functionally relevant.

Minor comments

- Page 10, line 223 and other places: TNFalpha, please fill in the Greek symbol alpha.
- Figure 2: it would be easier to understand the legend of the figure if it described what is seen in the figure, e.g., to shortly explain the terms CML exposed versus CML. Moreover, there is a typo: "Ph- cells progenitor cells" in the title of the legend.

Reviewer #2 (Remarks to the Author):

In this manuscript by Dawson et al. the authors have explored the bone marrow microenvironment in chronic myeloid leukemia (CML) with a special focus on macrophages. The authors use sophisticated technologies such as single cell RNA sequencing, metabolomics and secretory proteomics and the well-characterized transgenic murine model of CML to show that the CML microenvironment harbors unique populations of inflammatory and anti-inflammatory macrophages. The authors also demonstrate that

these macrophages from a CML environment express the markers CD36 and TGFbeta1, which distinguish them from macrophages from a normal environment. The authors show the partial responsibility of lactotransferrin, secreted by these CML macrophages, for some of the observed effects like suppressed phagocytosis etc.

The manuscript is of interest, but it approaches the problem in a somewhat simplistic way and fails to mention the very controversial role of macrophages in normal hematopoiesis and does not cite recently published papers on monocytes/macrophages in the leukemic environment. To a very large extent the manuscript relies on descriptive 'omics' without proving functionality in follow up experiments, e.g. if the NF-kB is found to be activated, it is essential to prove that in a simple Western blot +/- inhibitors.

1. In the characterization of the mouse model (which is well established and does not really need that much introduction), it would be important - for completeness's sake - to provide further information like WBC count, platelets, spleen weight and survival.
2. Which population amongst the C1-C6 population represents M1 or M2 macrophages? Changes in these cell populations during treatment with a TKI would be of interest.
3. Would CD86 upregulation correlate with the success of immunotherapy or treatment-free remission? This should be discussed.
4. LTF has been previously described as a tumor suppressor, while here it appears to have a different role. This needs to be discussed.
5. The paper overall lacks functional assays, coculture experiments etc. For example, if the expression of CD86 is altered, it would be important to see if CML conditioned medium or LTF as supplement may change T-cell activation by macrophages in a co-culture system.
6. Also, does administration of LTF to mice with CML alter the disease burden or survival?
7. Another molecule that needs more discussion and experimentation is CD36. Intriguingly, CD36 expression is decreased in CML macrophages. A previous publication (PMID: 27374788) has shown that the number of CD36+ LSC increases during CML progression in the murine model used. Did the expression of CD36 on leukemia cells change during CML progression in the mentioned murine model, especially after chemotherapy treatment? A discussion about the diverse role of CD36, including fatty acid metabolism will be important.
8. What is the consequence of altered metabolism in CML macrophages?
9. F4/80+ CD11b+ as markers are limited in describing macrophages. Better markers are

needed, such as CD169 etc. Dendritic cells and others may also be positive for F4/80+ CD11b+.

10. Some of the findings presented here are also shown in the cited papers by Welner et al and Schepers et al. and some of the same findings in this paper are shown by different methods. Other findings like reduced adhesion of CML cells have been known for many years.

11. Tgfbeta1 is a very promiscuous protein whose effects are varied. Therefore, any possible overstatements, e.g. line 201, should be avoided.

12. Do the Lin- c-Kit+ cells remain the same cell type after culture?

13. The number of mice per experiment seems quite low in some experiments.

14. What exactly is driving decreased phagocytosis? Knockout studies are needed here.

15. The effect of irradiation on CD45.1+ non-CML macrophages must not be overlooked and may be influencing the results.

16. Discussion: Please discuss other functions of CD36. The discussion in general should be a more critical evaluation of the authors' compared to other authors' work.

17. How did the authors arrive at the concentration of 50-200 ug of LTF? This seems rather high.

18. What is the phenotype of BMDM after exposure to M-CSF and culture for 4 days?

19. Figure 2a: Why do CD45.1+ LK cells increase after CML exposure?

Minor:

There are some issues with spelling, syntax and grammar.

Reviewer #3 (Remarks to the Author):

Reviewer Comments for the Authors:

Dawson et al used a chimeric mouse model of Chronic Myeloid Leukemia to characterize the role macrophages play in this disease, specifically how macrophages are altered functionally and transcriptionally. Little is known about macrophage function in CML, and here, the authors show that the CML microenvironment alters macrophage immune effector function.

The authors chimeric mouse model demonstrated myeloid bias, consistent with CML. It is revealed that macrophage phagocytic function is suppressed, which is supported by gene expression analysis of phagocytic related genes and phagocytic activity assays. The single cell RNA seq showed two subpopulations of immature and anti-inflammatory macrophages, in which both populations report low expression of phagocytic related genes. Pathway analysis on these two populations show that CML macrophages have suppressed inflammatory and phagocytic gene signatures.

The paper is detailed and well written. However, in its present form it is largely descriptive and the relevance of the reported changes in macrophage function to CML pathogenesis (human or murine) or disease response are unclear. The relevance of these findings would be supported by showing 1) that depth of response in human CML or ability to achieve TFR is correlated with macrophage function (or any relevant clinical outcome), 2) that perturbations in macrophage function alter treatment responses to TKI in murine CML. Without such evidence, the relevance of this work is speculative which reduces its impact.

Specific Comments

In the introduction (line 77), the authors should include brief background information about the current research done on macrophages in CML.

It would be clearer to update the heatmaps in Figures 3A and 4B with a color range that can be more easily visualized. In the present form, the heatmap show little dynamic range and appear either red or blue.

The authors should include a sample plot of the flow cytometry data that is represented as bar

Figure 1B-C. The authors should either add an extra key to the stacked bar graph or create a double bar graph. The double bar graph can come across as confusing.

Figure 2: It would be helpful if the authors included some long-term mouse viability data and/or periodic bleeding data overtime (i.e changes in macrophage % overtime). It would also be helpful if the graphs were further condensed.

Line 134/Figure 3. Confirm downregulation of macrophage genes CD36 and lamp2 with qPCR or western blot in CD45.1+CD11b+F4/80+ macrophages (ignore comment if its in the supplement)

Line 144-147 / 3e. The authors state that CML conditioned media resulted in a significant downregulation of BMDM phagocytosis, however this change is subtle and should be noted.

Line 278 / Figure 6e/f . The follow up on the secreted proteome is rather limited and does not completely explain the observed phenotypic differences in bone marrow macrophages. It is not clear why the authors did not evaluate other candidates identified in their screen.

Line 275 Related to the secreted proteome, the authors state the PCA revealed “clear separation between control and CML mice.” The supplied PCA shows minimal separation to my eye, suggesting either a small effect size or high degree of technical noise in this experiment. The authors should moderate their claims and or consider repeating the experiment to improve the quality.

Line 125-126 “thus demonstrating that the CML niche supports myelopoiesis of the BCR-ABL negative population” This statement is not justified by the data. There is an expansion of BCR-ABL1 negative myeloid cells, but the authors do not provide any data supporting this being niche-derived.

Line 200-201 “Here we demonstrate that the CML BM niche supports an immune suppressive environment by driving high expression of Tgfbi and Lgals1.” This is an overstatement of the findings. It would be more correct to say that the CML BM niche results in decreased macrophage function which is associated high expression of tgfb1 and Lgals1. The statement above suggests causation but data is not provided to support this.

Line 345-347 “Future work should aim to investigate if cd36 expression can be used to monitor macrophage immune function and homeostasis in CML progression or during TKI treatment.” Murine and human cell surface markers often do not have 1:1 correspondence. In order for this to be a future direction, CD36 would have to mark this population in both mouse and human systems. Can the authors provide evidence to support this?

Summary: We would like to thank all the Reviewers for their constructive comments, and we believe that by addressing them our manuscript has substantially improved.

Initially, to confirm that macrophages play a critical role in our mouse model, we demonstrate that temporal depletion of macrophages prior to leukaemia induction (using CSF1R antagonist antibody), increases leukaemia burden and decreases survival of chimeric mice following BCR-ABL induction (**new Fig. 2a-e**).

Mechanistically, we now provide convincing evidence that Ph⁻ bone marrow (BM) macrophages exposed to a CML microenvironment display reduced efferocytosis (clearing of apoptotic cells) and phagocytic capacity (**new Figs. 3a-c; 5a-b**). Treatment of primary mouse and human BM derived macrophages (BMDM) with the CD36 inhibitor sulfosuccinimidyl oleate, or genetic deletion of CD36 in THP-1 derived macrophages, results in reduced clearance of apoptotic cells *in vitro* (**new Figs. 3e-f; 6a-c, f; Suppl. Fig. 15a-b**). Notably, while the diminished efferocytosis is mediated by reduced expression of CD36, the reduction in phagocytosis occurs independently of CD36 (**new Suppl. Fig. 15c**). Additionally, treatment with LTF reduces CD36 transcript and protein levels on BMDM and/or THP-1 derived macrophages and replicates the effect of CML exposure on BMDM, including decreased phagocytic function of mouse BMDM and THP-1 derived macrophages (**new Figs. 5g-h; 6c, e**). Furthermore, mouse/human BMDM display reduced clearance of apoptotic CML cells following LTF exposure, strengthening our message that secreted LTF is a key contributor to the immature phenotype of CML exposed macrophages (**new Fig. 6a-b, f**).

To further increase the clinical relevance of our study, we established new collaboration with Professor Tiong Lab, who have recently performed scRNA-seq on BM samples obtained from CML patients and healthy donors. Applying this unique dataset, we reveal that macrophages from individuals with CML are transcriptionally distant from macrophages obtained from healthy BM (**new Supp. Fig. S9a-f**). We also uncover that, in contrast to Ph⁻ CML exposed mouse macrophages, human BM macrophages have pro-inflammatory signature, which is not unexpected given that the majority of macrophages in the BM of CML patients at diagnosis are likely to express the BCR-ABL onco-protein.

Below we provide point-by-point responses to all three Reviewers.

Reviewer #1 (Remarks to the Author):

The manuscript presents experimental work using state-of-the-art methods studying the role of macrophages in CML on a single cell level. The authors used a chimeric CML mouse model to study the role CML cells on the marrow microenvironment and on local bystander macrophages. The major findings of the paper were:

- CML cells can alter macrophages transcriptomes and phagocytic function.
- CML exposed macrophages show down-regulation of phagocytosis genes CD36 and Lamp2.
- CML cells secrete factors including lactotransferrin, which suppress phagocytosis, mitochondrial respiration, and inflammatory genes.

Response: We thank Reviewer 1 for commending on the state-of-the-art methods we have used and for nicely summarising our main findings.

Major concerns:

There are several major concerns with the manuscript:

1. All experiments were carried out with an (elegant) murine model or with murine cell lines. Therefore, the relevance of the findings for human disease remains unanswered. It would be highly desirable to conduct several experiments with human cells or tissues to create evidence for relevance in human CML.

Response: We have made huge effort to address this important comment and believe that the relevance to human CML has now substantially increased. Significantly, we optimised methods for human macrophage differentiation of myeloid cells derived from the femoral head BM from individuals undergoing hip replacement. We applied these primary normal (Ph⁻) macrophages and assessed efferocytosis following *in vitro* culture with conditional medium obtained from primary human CML CD34⁺ cells (leukaemia conditional medium; LCM) and normal counterparts (conditional medium; CM). These data demonstrate that treatment with the sulfosuccinimidyl oleate (SSO; an irreversible CD36 inhibitor), or lactotransferrin treatment, reduces efferocytosis in Ph⁻ BM macrophages to similar degree as treatment with CML conditional media (new Fig. 6a).

Additionally, to provide further mechanistic insight and complement work on mouse/human BM derived macrophages, we reveal that pharmacological CD36 inhibition and CD36 knockdown reduces efferocytosis of THP-1 derived macrophages, with similar effect seen following lactotransferrin treatment (new Fig. 6b, f; Suppl. Fig. S15a-b). We confirm in THP-1 derived macrophages that CML conditional media (from human CML cells) leads to downregulation of CD36 expression, with similar effect seen following lactotransferrin treatment (new Fig. 6d-e). Moreover, SSO treatment, lactotransferrin treatment or CD36 knockdown reduces CD11b expression in THP-1 derived macrophages, suggesting reduced differentiation (new Fig. 6g-h). Notably, we also uncover that

lactotransferrin treatment reduces phagocytosis in THP-1 derived macrophages through a mechanism that is independent of its effect on CD36 expression (**new Suppl. Fig. S15c**).

Finally, we have interrogated relevant scRNA-seq dataset, which demonstrated that human BM macrophages, isolated from CML patients, are transcriptionally distinct and have inflammatory signature when compared with macrophages obtained from the BM of healthy donors (**new Suppl. Fig. S9a-f**). Although it is most likely that this represents a signature of Ph⁺ macrophages, as far as we know, this is the only available dataset on human BM derived macrophages from CML patients.

Relevant text has been added on Pages 11-16 in the revised manuscript (highlighted in Red).

2. Overall, the manuscript remains descriptive in large parts. For example, no mechanistic explanation is given on the regulatory molecules or signaling pathways that are induced in murine macrophages when exposed to CML cells or secreted factors (such as lactoferrin). No mechanistic explanation is given regarding the downregulation of some potentially relevant molecules such as CD36 or Lamp2.

Response: As mentioned above, we have performed additional experiments and provide new robust mechanistic data regarding the effect of lactotransferrin on efferocytosis (CD36 dependent) and phagocytosis (CD36 independent) using mouse and both human primary and THP-1 derived macrophages. We also confirm that while the CD36 reduction following macrophage culture in CML conditional media is likely, at least in part, mediated by lactotransferrin, the reduction in LAMP2 expression is mediated independently of increased lactotransferrin exposure (**new Fig. 5g-h**). Note, Figure 5g-h has been performed by treating mouse BM macrophages when cultured in control conditional media, which we believe is more relevant than our previous experiment where the macrophages were cultured in “macrophage” medium.

To investigate the potential downstream signaling pathway(s) involved we speculated that CML cell exposure would downregulate NF- κ B signalling (main regulator of inflammation). This was guided by our GSEA analysis showing reduced “*TNF α signalling via NF- κ B*” in CML exposed macrophage clusters, and the significant reduction in TNF α expression in BM derived macrophages following exposure to CML conditional medium (**Suppl. Fig. S7a-d**). However, we did not observe significant changes in Serine536 phosphorylation on the p65 NF- κ B subunit (which controls the activation and kinetics of NF- κ B p65 nuclear import) following 24hr exposure to CML conditional medium (**new Suppl. Fig. S7e**). It is possible that the reduction in TNF α expression might require longer time to mediate its effect on NF- κ B activity, so this part still requires more detailed investigation in follow-up studies.

Overall, while we acknowledge that more work is needed to precisely answer how CML conditional media/lactotransferrin treatment reduces phagocytosis and CD36-mediated efferocytosis and the downstream signalling pathway involved, we strongly believe the new data are highly relevant, novel and have significantly improved our manuscript.

Relevant text has been added on Pages 10 and 15.

3. Similarly, on a functional level, the manuscript could benefit from functional experiments confirming the findings by alternative approaches. For example, the claim (e.g., Figure 3a) that the reduced expression of CD36 is functionally relevant for the phagocytic function of macrophages in the bone marrow of CML could be tested by blocking of CD36 in CD36 positive macrophages or CD36 gene deletion experiments. These experiments could be carried out, e.g., with coculture of CML cells or in functional tests of phagocytosis.

Response: We appreciate this constructive comment and valuable suggestions. We have now addressed this experimentally and provide robust mechanistic data demonstrating that treatment of mouse and human (both primary BM derived, and THP-1 derived) macrophages with CML conditional medium reduces phagocytosis, efferocytosis and CD36 expression (new Figs. 3a-d; 5a-b, g-h, 6a-e; Suppl. Fig. S15a-b). Similar effect is seen on efferocytosis following genetic or pharmacological inhibition of CD36, or lactotransferrin treatment, while the effect on phagocytosis occurs in CD36 independent way (new Figs. 5a-b; 6a-f; Suppl. Fig. S15c). Overall, we believe our new mechanistic data has substantially increased the focus of our work and will be of significant interest to the field.

Relevant text has been added on Pages 11-16.

Additional, major points:

Figure 2e: There is an apparent difference regarding the number of CD8+ T cells in the CML compartment versus the CML exposed compartment. How do the authors explain this difference?

Response: While it is true that BCR-ABL does not alter the relative frequency of CD8+ T cells (Fig. 1e; Rebuttal Fig. 1a), it significantly increases the absolute number of CD8+ T cells (Rebuttal Fig. 1b). Notably, CD8+ T cells are significantly reduced in non-transformed CD45.1 BM (Fig. 1e; Rebuttal Fig. 1c-d). As the reviewer suggests, there is indeed a difference in CD8+ T cell number in the CML compartment versus the CML exposed compartment. Interestingly, Schürch and colleagues (PMID: 23401488) demonstrated that cytotoxic T cells promote LSC expansion by the production of effector cytokines such as IFN γ . Thus, it is possible that BCR-ABL drives expansion of Ph⁺ cytotoxic cells to promote survival of LSCs.

Rebuttal Figure 1

Figure 4a and b:

Can any of the clusters C1-C6 be found in human CML? It would be desirable to expose human macrophages to human CML cells (or supernatant) to investigate the relevance of the findings for human disease.

Response: To precisely address if clusters C1-C6 exist in individuals with CML would require generation of/access to single cell transcriptomic dataset from BM of CML patients, accompanied with DNA sequencing to identify Ph⁻ macrophages. As far as we know, such dataset does not yet exist. However, to inform whether macrophages in the CML BM are transcriptionally altered in humans, we established new collaboration with Professor Tiong Lab, who have recently performed single cell RNA-seq on BM mononuclear cells, isolated from healthy donors and individuals with chronic phase CML (at diagnosis). By applying this unique dataset, we demonstrate that BM macrophages from CML patients are transcriptionally different from BM macrophages from healthy donors (cluster separately), with increased expression of inflammatory markers such as *NF-kB1A*, *TNF*, *CXCL8*, *CCL4*, *CCL3* and *CXCL2* (new Suppl. Fig. S9a-f). However, while we believe this is of interest to the field, the limitation of this dataset is that we cannot specifically identify Ph⁻ macrophages, so it is highly likely that the inflammatory signature is driven by BCR-ABL expression. Additionally, it is very likely that that successful TKI treatment targets Ph⁺ macrophages in the BM, so it is unclear whether this signature persists following TKI treatment and should be investigated in follow-up studies.

Regarding exposing human macrophages to human cells, we have performed additional experiments using primary BM derived macrophages and THP-1 macrophages and exposed them to conditional medium from primary normal and CML CD34⁺ cells (new Fig. 6a-h). In short, this has confirmed our results using murine BM derived macrophages and provided important relevance to human disease.

Relevant added text is on Pages 11 and 15-16.

Figure 7a: could these experiments be reproduced with human macrophages or macrophage cell lines?

Response: This has now been done and addressed in our previous comments.

Figure 7g: In this figure, one can see significant changes only for TNFalpha and TGFbeta but not CD36, LAMP2 or others. This represents a contradiction to other experiments, e.g., of Figure 4d. This means that LTF only produces a subset of the changes seen with CML exposed cells. What other proteins of the secretome contribute to the observed effects?

Response: As briefly mentioned in response to comment 2, we have repeated this experiment but cultured the macrophages in conditional medium (+/- lactotransferrin) instead of using "macrophage medium" as we did previously (Figure 7g has now been replaced with new data). We believe this is a more logical experimental setup as it accounts for other contributing factors that may be secreted from stem/progenitor cells. This revealed that lactotransferrin, when "spiked" into conditional medium, results in a significant reduction in CD36 expression, while no significant changes were observed for

Lamp2, *TNF α* or *Tgf β* (new Fig. 5g-h – which can be compared with new Suppl. Fig. S14 where lactotransferrin is added to BM derived macrophages in “macrophage medium”). Overall, while we agree that lactotransferrin is not the sole contributor to the observed phenotype, we have now provided strong evidence that it is an effective modulator of macrophage function. What effect the other secreted proteins have on macrophage function warrants further investigation and is beyond the focus of this manuscript.

Relevant added text is on Pages 14-15.

Figure 7h: The oxygen consumption rate shown upon 24-hr LTF stimulation drops somewhat, but this seems a rather small change. Please comment whether these changes are functionally relevant.

Response: We agree with this comment and acknowledge that the modest reduction in mitochondrial respiration may not be a major factor in mediating the functional changes we observe in macrophages. Nonetheless, we have performed additional experiments in human THP-1 derived macrophages, following CD36 knockout, pharmacological CD36 inhibition and lactotransferrin treatment. Here we observed similar reduction in oxygen consumption rate with CD36 knockout and lactotransferrin treatment, with lactotransferrin having no effect in CD36 depleted cells (new Suppl. Fig. 15d-e). However, since the changes are relatively modest and relevance of these metabolic changes are still unclear, we have moved the original Figure 7h to Supplemental file (Suppl. Fig. 14h), presented new data as supplementary figure and toned down the interpretation of these findings.

Relevant added text is on Page 16.

For the interest of this Reviewer, these data made us hypothesize that CD36 is involved in fatty acid transport, followed by fatty acid oxidation in the mitochondria (and therefore contributing to mitochondrial respiration), in macrophages. However, using heavy labelled palmitate, we were unable to see significant labelling into TCA-cycle metabolites in THP-1 cells, preventing us to fully address this hypothesis.

Minor comments

- Page 10, line 223 and other places: TNF α , please fill in the Greek symbol alpha.

Response: This has now been corrected.

- Figure 2: it would be easier to understand the legend of the figure if it described what is seen in the figure, e.g., to shortly explain the terms CML exposed versus CML. Moreover, there is a typo: “Ph- cells progenitor cells” in the title of the legend.

Response: This has now been explained in the figure legend (text below) and typo has been corrected.

Relevant added text in Figure legends: *“CML exposed cells refer to CD45.1 WT BM cells exposed to CD45.2 SCLtTA+/BCR::ABL1⁺ (CML) BM cells. CML exposed cells have been compared to CD45.1 WT BM cells exposed to CD45.2 SCLtTA+/BCR::ABL1⁻ (Control)”*

Reviewer #2 (Remarks to the Author):

In this manuscript by Dawson *et al.* the authors have explored the bone marrow microenvironment in chronic myeloid leukemia (CML) with a special focus on macrophages. The authors use sophisticated technologies such as single cell RNA sequencing, metabolomics and secretory proteomics and the well-characterized transgenic murine model of CML to show that the CML microenvironment harbors unique populations of inflammatory and anti-inflammatory macrophages. The authors also demonstrate that these macrophages from a CML environment express the markers CD36 and TGFbeta1, which distinguish them from macrophages from a normal environment. The authors show the partial responsibility of lactotransferrin, secreted by these CML macrophages, for some of the observed effects like suppressed phagocytosis etc.

The manuscript is of interest, but it approaches the problem in a somewhat simplistic way and fails to mention the very controversial role of macrophages in normal hematopoiesis and does not cite recently published papers on monocytes/macrophages in the leukemic environment. To a very large extent the manuscript relies on descriptive 'omics' without proving functionality in follow up experiments, e.g. if the NF-kB is found to be activated, it is essential to prove that in a simple Western blot +/- inhibitors.

Response: We value that Reviewer 2 acknowledges the potential interest of our work and appreciate the constructive comments. Overall, as mentioned in our summary and responses to Reviewer 1, comments 1-3, we believe we have now significantly improved our manuscript by providing mechanistic insight of how CML cells may affect the function of BM macrophages. In short, we provide robust data demonstrating that CML exposed macrophages (both mouse and human) have reduced efferocytosis and phagocytosis potential when compared with control macrophages (new Figs. 3 and 5; Suppl. Fig. S10). Moreover, we show that CML exposure reduces the levels of CD36, which is also seen following lactotransferrin treatment. Mechanistically, using genetic and pharmacological approaches, we uncover that lactotransferrin treatment is sufficient to reduce efferocytosis and phagocytosis capacity of BM macrophages, and while the effect on efferocytosis is dependent on CD36 reduction, the effect on phagocytosis of latex beads is independent of CD36 expression (new Fig 6; Suppl. Fig. S15).

As also mentioned in response to Reviewer 1, comment 2, we have performed additional experiments to further investigate the potential downstream signalling changes following CML exposure. We hypothesised that CML cell exposure would downregulate NF-kB signalling (main regulator of inflammation). This was guided by our GSEA analysis showing reduced "*TNF α signalling via NF-kB*" in CML exposed macrophage clusters, and the significant reduction in TNF α expression in BM derived macrophages following exposure to CML conditional medium (Suppl. Fig. S7a-d). However, we did not observe significant changes in Serine536 phosphorylation on the p65 NF-kB subunit (which controls the activation and kinetics of NF-kB p65 nuclear import) following 24hr exposure to CML conditional medium (new Supp. Fig. S7e). It is possible that the reduction in TNF α

expression might require longer time to mediate its effect on NF- κ B activity, so this part still requires more detailed investigation in follow-up studies.

Relevant added text is on Pages 10 and 12-16.

1. In the characterization of the mouse model (which is well established and does not really need that much introduction), it would be important - for completeness's sake - to provide further information like WBC count, platelets, spleen weight and survival.

Response: We have extensive experience with this model and routinely use the increase in GR1⁺CD11b⁺ myeloid cells in blood and BM as an indicator of leukaemia (which we have done in Fig. 1 and new Fig. 2). In Fig. 4 we also included whole blood counts and spleen weights (Suppl. Fig. 8a and 10a in first version). However, in the interest of space (revised manuscript includes 15 Supplementary Figures) and given that this is a well characterised model, we have removed these data from the manuscript. Survival data are now provided in new Fig. 2 following macrophage depletion.

2. Which population amongst the C1-C6 population represents M1 or M2 macrophages? Changes in these cell populations during treatment with a TKI would be of interest.

Response: The concept of classically activated macrophages (M1; produce pro-inflammatory cytokines) and alternatively activated (M2) macrophages, relies on stimulating macrophages *in vitro* with a defined set of factors. A limitation to this perspective is that macrophages, taken out of their native environment and stimulated *in vitro*, change dramatically. Additionally, macrophages are known to be versatile cells with functional plasticity, so placing macrophages isolated directly from the BM microenvironment into opposite end of this spectrum is expected to be challenging. That said, following characterisation of our CML exposed macrophages (C2 and C6), the top marker genes driving the unsupervised clustering included *Tgfbi*, *Lgals1* and *Ly6c2*, which have been related to alternatively activated (M2-like) macrophages (Fig. 2h-j; Suppl. Fig. S6a-b). However, our data strongly suggests that CML exposure is associated with Ph⁻ immature macrophage populations and given that many leaders in the immuno-oncology field have discouraged the use of this reductive M1/M2 model as it might restrain, rather than enable discovery, we have deliberately avoided using the M1/M2 classification.

We agree that changes in the clustering during treatment would be of interest but strongly believe that it is beyond the focus of current work. Indeed, we have now secured funding to follow up on this work, with one objective in that grant aiming to address this, which will require additional *in vivo* experiment and single cell RNA-seq during and following TKI treatment.

3. Would CD86 upregulation correlate with the success of immunotherapy or treatment-free remission? This should be discussed.

Response: Our data shows that CML conditional media, pharmacological inhibition of CD36, and LTF treatment leads to upregulation of CD86 (new Figs. 3g; 5c). On one hand, CD86

provides essential signals for T cell activation and cytokine production by binding to CD28. On the other hand, CD86 can bind CTLA-4, a coinhibitory molecule that is induced on activated T cells leading to T cell deactivation and termination of immune response. It is possible that high-CD86 expressing macrophages inhibit T cell activation, and therefore, treatment with CTLA-4 specific antibody would have anti-leukaemia effect. However, although an interesting concept, to our knowledge this has not been tested, and whether CD86 upregulation correlates with sustained TFR also remains an open question.

4. LTF has been previously described as a tumor suppressor, while here it appears to have a different role. This needs to be discussed.

Response: It is true that LTF has been described to act as a tumour suppressor by regulating cell proliferation in several solid tumours (PMID: 10412049; PMID: 23069661). It has also been demonstrated that LTF is downregulated in human breast carcinoma cells (PMID: 1733438). However, in our model, LTF is significantly upregulated in CML LT-HSCs (**Suppl. Fig. S13**). Furthermore, secretion of LTF impairs the function and maturation of Ph-macrophages, suggesting that LTF acts a tumour promoter in our models. Whether LTF affects LSC function and proliferation in an autocrine manner needs further investigation.

Relevant text has been added in Discussion section, Page 20: *“Of note, LTF has been described to act as a tumour suppressor in several models of solid tumours including breast and nasopharyngeal carcinomas^{35, 36}. However, in our model LTF is likely to work as a tumour promoter by impairing the maturation and function of Ph- macrophages. The autocrine effect of LTF on LSCs requires further investigation.”*

5. The paper overall lacks functional assays, coculture experiments etc. For example, if the expression of CD86 is altered, it would be important to see if CML conditioned medium or LTF as supplement may change T-cell activation by macrophages in a co-culture system.

Response: We agree that this would be informative experiment. However, given that Reviewer 1 suggested more mechanistic data regarding the function of CD36 and the specific effect of lactotransferrin, we have focused more on that part (see comments 2-3 to Reviewer 1 above).

That said, we did additionally perform a challenging experiment to specifically assess the effect of CML conditional medium on activation of ovalbumin (OVA) antigen specific T cell response. Here, lymph nodes and spleens were dissected from OT-II transgenic mice. CD3⁺ CD4⁺ T cells were FACS sorted and isolated T cells were confirmed to express the transgenic ovalbumin specific TCR (V α 2V β 5) by flow cytometry. Murine BM derived macrophages were loaded with OVA protein and cultured with CTV stained CD4⁺ T cells for 72hr in the presence of control or CML conditional medium. The expression of T cell activation markers was then assessed at 72hr hours, measuring the expression of cell surface CD62L and CD44. Proliferation was also assessed by measuring CTV levels at time 0 and 72hr. However, we found that CML exposure had no effect on activation of CD4⁺ T cells at 72hr exposure to OVA presenting macrophages. A small reduction in the expression of CD62L on CML

exposed CD4⁺ T cells was observed but this was not statistically significant. Finally, we observed no changes to T cell activation in the presence of CML conditional medium. Therefore, we are still unable to confirm that the phenotypic or functional changes in CML exposed macrophages (or lactotransferrin supplementation) promotes changes in T cell activation. We aim to investigate this further in follow-up studies.

6. Also, does administration of LTF to mice with CML alter the disease burden or survival?

Response: Given that LTF addition to conditional medium reduces efferocytosis and phagocytosis in mouse and human macrophages (**new Figs. 5a-b; 6a-c**) our hypothesis is that LTF administration would exaggerate the disease (and we should aim to prevent LTF secretion from CML stem/progenitor cells instead). We are also unsure how LTF administration would impact overall the immune infiltrate, as it has been shown to affect other immune populations like dendritic cells (DC) (PMID: 29966271).

As an alternative, and a more specific approach, we have focused our effort in providing clearer evidence regarding the protective role of BM macrophages in CML. To achieve this, we depleted macrophages using CSF1R antibody and measured survival following leukaemia induction. This revealed that animal with reduced pool of BM macrophages had significantly reduced survival when compared with animals with intact macrophage pool (**new Fig. 2a-d**).

Relevant text has been added on Page 7.

7. Another molecule that needs more discussion and experimentation is CD36. Intriguingly, CD36 expression is decreased in CML macrophages. A previous publication (PMID: 27374788) has shown that the number of CD36⁺ LSC increases during CML progression in the murine model used. Did the expression of CD36 on leukemia cells change during CML progression in the mentioned murine model, especially after chemotherapy treatment? A discussion about the diverse role of CD36, including fatty acid metabolism will be important.

Response: We agree with the Reviewer that the effect we have seen on CD36 levels in CML exposed Ph⁻ macrophages is intriguing. Moreover, the fact that CD36 is the most significantly upregulated gene in human CML stem cells (CD34⁺38⁻) when compared to normal CD34⁺38⁻ cells (**Rebuttal Figure 2**), it is perhaps not surprising that CD36 levels are not reduced in Ph⁺ macrophages (**new Suppl. Fig. 4b** and data not shown).

Although we have not assessed CD36 levels in the leukaemic clone during leukaemogenesis or following TKI treatment in current work, we have substantially expanded our investigation on the role of CD36 and provide new mechanistic data showing that CML conditional media (or LTF treatment) reduced efferocytosis in a CD36 dependent manner, and phagocytosis in a CD36 independent manner, in human macrophages (**new Fig. 6f; Suppl. Fig. S15c**). See also Responses to Reviewer 1, comments 2-3.

Regarding the role of CD36 in fatty acid oxidation, as mentioned in Response to Reviewer 1 (comment regarding Figure 7h), the decreased expression of CD36 in CML exposed macrophages, coupled with a decrease in mitochondrial respiration, we hypothesized that CD36 is involved in fatty acid transport, followed by fatty acid oxidation in the mitochondria (and therefore contributing to mitochondrial respiration), in macrophages. However, using heavy labelled palmitate, we were unable to see significant labelling into TCA-cycle metabolites in THP-1 cells, preventing us to fully address this hypothesis.

8. What is the consequence of altered metabolism in CML macrophages?

Response: As mentioned above, we have tried to provide more mechanistic data regarding the specific role of altered metabolism in CML macrophages. We have revealed that, in addition to co-culture with CML stem/progenitor cells or LTF treatment, CD36 knockout results in modest reduction in basal and ATP-linked mitochondrial respiration in human THP-1 derived macrophages (**new Suppl. Fig. S15 a-b, d-e**). However, we were unable to provide robust evidence that this was due to a reduction in fatty acid oxidation. Therefore, we have presented all data regarding altered metabolism as supplementary files and toned-down relevant text in result/discussion section.

It is tempting to speculate that there is a mechanistic connection between reduced metabolic activity, the reduction in efferocytosis/phagocytosis and more immature stage of macrophages. However, to test this would require direct modulation of metabolic activity, followed by further assessment of efferocytosis/phagocytosis/maturation. Although we agree that these experiments, and further investigation into the consequence of altered metabolism in CML exposed macrophages would be interesting, we were reluctant to focus on this part given the relatively modest reduction we observed in our initial experiments.

9. F4/80+ CD11b+ as markers are limited in describing macrophages. Better markers are needed, such as CD169 etc. Dendritic cells and others may also be positive for F4/80+ CD11b+.

Response: We acknowledge that DC can express macrophage markers, with conventional DC (cDC) and DC-like subsets expressing CD11b. However, it has been shown that cDC mature in peripheral tissue, while only pre-cDC are found in the BM (PMID: 28198365). Nevertheless, in subsequent experiments we included CD11c in our macrophage/monocyte panel to exclude cDC. While in the spleen we have an apparent population of cDC, this population is much smaller in the BM (**Rebuttal Figure 3**). To gate for macrophages, we excluded the CD11c⁺ population. Macrophages were then defined as F4/80⁺ CD11b⁺.

Rebuttal Figure 3

10. Some of the findings presented here are also shown in the cited papers by Welner et al and Schepers et al. and some of the same findings in this paper are shown by different methods. Other findings like reduced adhesion of CML cells have been known for many years.

Response: We agree and were pleased that some of our findings agreed with previously published results by these groups and have referenced their work. Given that we have now added significant amount of new mechanistic data regarding efferocytosis, phagocytosis and the link with LTF mediated changes in CD36 levels, we have merged Figures 1 and 2 in our previous version and replaced some panels into Supplementary file. We believe this has resulted in a manuscript with increased flow and more focus on novel findings. We have not focused our work on reduced adhesion or location of CML cells and therefore not discussed this in detail.

11. Tgfbeta1 is a very promiscuous protein whose effects are varied. Therefore, any possible overstatements, e.g. line 201, should be avoided.

Response: The wording has now changed.

New text on Page 11: *“Overall, we demonstrate that the CML BM niche drives high expression of genes related to alternatively activated macrophages such as Tgfb1 and Lgals1.”*

12. Do the Lin⁻ c-Kit⁺ cells remain the same cell type after culture?

Response: To answer this, c-kit enriched BM was stained to determine frequency of LK, LSK, MPP and LT-HSC population at baseline (0hr) and following 48hr culture. As can be seen in the figure below (**Rebuttal Figure 4**), Lin⁻ c-Kit⁺ persist following a 48hr incubation, while there is a reduction of the most primitive LT-HSC population, suggesting that the most primitive cells undergo differentiation.

Rebuttal Figure 4
13. The number of mice per experiment seems quite low in some experiments.

Response: We agree that increasing the number of mice (we have used n=4 in most cases) would give us more statistical power. However, taking 3R into account and guidance from previously published work, we believe that our experimental setup was appropriate.

14. What exactly is driving decreased phagocytosis? Knockout studies are needed here.

Response: We have put a lot of effort into further investigating the mechanism underlying decreased phagocytosis in CML exposed macrophages. Here I would like to refer to our Responses to Reviewer 1, comments 1-3. Specifically, we show that CML exposure and/or LTF treatment, reduces CD36 expression, efferocytosis and phagocytosis of latex beads in mouse and human macrophages (**new Figs. 3a-d; 5a-b, g-h; 6a-e**). Importantly, through CD36 knockout studies we reveal the LTF mediated reduction in efferocytosis is a result of reduced CD36 expression, whereas the effect on phagocytosis is independent of CD36 levels (**new Fig. 6f; Suppl. Fig. 15a-c**). Note, the reduction in LAMP2 levels following CML exposure is not reproduced by LTF treatment (**new Fig. 5g**).

Relevant text has been added on Pages 12-16.

15. The effect of irradiation on CD45.1+ non-CML macrophages must not be overlooked and may be influencing the results.

Response: We agree with this point and have tried to take this into account within our experimental setup. As mentioned in the Method section (Mouse models) *“To allow for irradiation recovery, mice were maintained on tetracycline (0.5g/L) for 14 days prior to removal of tetracycline for transgene induction”*, which we think is an appropriate length of time to allow the BM to recover from the irradiation.

16. Discussion: Please discuss other functions of CD36. The discussion in general should be a more critical evaluation of the authors’ compared to other authors’ work.

Response: This has now been improved for both sections regarding CD36 and LTF.

Most relevant added text is in Discussion section on Pages 17 and 19 (highlighted in red).

17. How did the authors arrive at the concentration of 50-200 ug of LTF? This seems rather high.

Response: Following measurements of LTF in CML conditional medium by ELISA we observed that LTF is present at around 130-180 $\mu\text{g}/\text{ml}$ concentration. We used this as a guidance when we performed the initial experiments testing the effect of LFT on RAW 264.7 cells (these data have now been removed in the new version and replaced with data on BM derived mouse and human macrophages). However, as mentioned in our Response to Reviewer 1 (comment regarding Figure 7g) we have repeated all the LTF treatment experiment and instead of culturing the macrophages in “macrophage medium” as we did previously we have now cultured the macrophages in conditional medium +/- 50 $\mu\text{g}/\text{ml}$ LTF, which is commonly used concentration in the literature and approximately accounts for the difference between CM and LCM LTF concentrations. We believe this is a more logical experimental setup as it accounts for other contributing factors that may be secreted from stem/progenitor cells.

18. What is the phenotype of BMDM after exposure to M-CSF and culture for 4 days?

Response: Below are pictures of murine BMDM after exposure to 20ng/mL M-CSF for 4 days. As can be seen, fully mature macrophages can be observed on day 4.

Rebuttal Figure 5

19. Figure 2a: Why do CD45.1+ LK cells increase after CML exposure?

Response: These results agree with findings by Welner *et al.* (Cancer Cell, 2015), which demonstrated an increase in CML exposed LSK and granulocyte-monocyte progenitor (GMP) populations. However, given our focus on the macrophage population, we have not investigated further how the CML microenvironment supports bystander progenitor cells.

Minor:

There are some issues with spelling, syntax and grammar.

Response: We have now gone over the whole manuscript, corrected spelling mistakes, and improved grammar as suggested.

Reviewer #3 (Remarks to the Author):

Reviewer Comments for the Authors:

Dawson et al used a chimeric mouse model of Chronic Myeloid Leukemia to characterize the role macrophages play in this disease, specifically how macrophages are altered functionally and transcriptionally. Little is known about macrophage function in CML, and here, the authors show that the CML microenvironment alters macrophage immune effector function.

The authors chimeric mouse model demonstrated myeloid bias, consistent with CML. It is revealed that macrophage phagocytic function is suppressed, which is supported by gene expression analysis of phagocytic related genes and phagocytic activity assays. The single cell RNA seq showed two subpopulations of immature and anti-inflammatory macrophages, in which both populations report low expression of phagocytic related genes. Pathway analysis on these two populations show that CML macrophages have suppressed inflammatory and phagocytic gene signatures.

The paper is detailed and well written. However, in its present form it is largely descriptive and the relevance of the reported changes in macrophage function to CML pathogenesis (human or murine) or disease response are unclear. The relevance of these findings would be supported by showing 1) that depth of response in human CML or ability to achieve TFR is correlated with macrophage function (or any relevant clinical outcome), 2) that perturbations in macrophage function alter treatment responses to TKI in murine CML. Without such evidence, the relevance of this work is speculative which reduces its impact.

Response: We thank Reviewer 3 for describing our manuscript as detailed and well written and for nicely summarising our findings. We have now taken all Reviewers comments onboard and resubmitted significantly improved manuscript. The parts we have particularly strengthened relate to the mechanistic insight of how CML exposure alters efferocytosis and phagocytosis in mouse and human macrophages and would like to refer to our Responses to Reviewer 1, comments 1-3 for more details.

As also mentioned in our initial summary above, we now provide more clinical relevance by interrogating recently obtained scRNA-seq data on BM samples obtained from CML patients and healthy donors. Through this analysis, we reveal that macrophages from individuals with CML are transcriptionally distant from macrophages obtained from healthy BM (**new Supp. Fig. S9a-f**). We also uncover that human BM macrophages have pro-inflammatory signature, which is likely to be driven by the BCR-ABL onco-protein. In relation to **new Suppl. Fig. S9e**, I would like to mention that cluster 4 (C4), a CML cluster that separates from control macrophages is entirely made up of CML patient belonging to the “Treatment failure” group (Group C refers to “Treatment failure”: See **Rebuttal Figure 6** below and Krishnan *et al*, Blood 2023). However, since this signature is coming from a single patient (P605) out of n=6 patient samples in Group C, we have not included this data in the manuscript and suggest analysis of a larger patient cohort is required to draw firm conclusion regarding correlation between macrophage function and clinical outcome. Therefore, more relevant

dataset required to address specifically whether the “depth of response in human CML or ability to achieve TFR is correlated with macrophage function” is not available yet (as far as we know) and would require single cell RNA-seq on macrophages following TKI treatment/sustained DMR (we believe this would also increase the changes of identifying Ph⁺ macrophage populations).

Specific Comments

In the introduction (line 77), the authors should include brief background information about the current research done on macrophages in CML.

Response: We have now referenced recent relevant paper that highlights the importance of splenic red pulp macrophages in LSC maintenance (PMID: 36163264). However, to our knowledge, literature related to role of macrophages in CML is otherwise limited.

It would be clearer to update the heatmaps in Figures 3A and 4B with a color range that can be more easily visualized. In the present form, the heatmap show little dynamic range and appear either red or blue.

Response: We have now made changes as suggested (now presented as **Figs. 2g and 2i**).

The authors should include a sample plot of the flow cytometry data that is represented as bar

Response: We believe we have now significantly improved visual presentation of all main and Supplementary Figures and changed all figures to scatter plots with bars. If the reviewer is specifically referring to Fig. 1b-c (now presented as **Suppl. Fig. S1b-c**) and prefers bars with all data points (**Rebuttal Figure 7**) we would be happy to include that in the manuscript.

Figure 1B-C. The authors should either add an extra key to the stacked bar graph or create a double bar graph. The double bar graph can come across as confusing.

Response: We have now changed the colour scheme in this figure to make it clearer (now presented as **Suppl. Fig. S1b-c).**

Figure 2: It would be helpful if the authors included some long-term mouse viability data and/or periodic bleeding data overtime (i.e changes in macrophage % overtime). It would also be helpful if the graphs were further condensed.

Response: We agree that this would be informative. As briefly mentioned in our Response to Reviewer 2, comment 6, in terms of long-term mouse viability data we have performed additional experiment where we depleted macrophages using CSF1R antibody and measured survival following leukaemia induction. This revealed that animal with reduced pool of BM macrophages had significantly reduced survival when compared with animals with intact macrophage pool (new Fig. 2a-d**). Relevant text has been added on Page 7.**

Regarding longitudinal changes in macrophage populations during leukaemogenesis or TKI treatment we have now secured funding to follow up on this work, with one objective to track macrophage changes overtime and during/following TKI treatment (as briefly mentioned to our Response to Reviewer 2 comment 2).

We believe we have now significantly improved visual presentation of all main Figures and Supplementary Figures in the manuscript.

Line 134/Figure 3. Confirm downregulation of macrophage genes CD36 and lamp2 with qPCR or western blot in CD45.1+CD11b+F4/80+ macrophages (ignore comment if its in the supplement)

Response: Using flow cytometry and/or qPCR we present new supporting data showing levels of CD36 and Lamp2 in mouse or human macrophages following exposure to CML conditional medium or lactotransferrin treatment (new Figs. 3d; 5g-h; 6d-e; Suppl. Fig. S14g**).**

Line 144-147 / 3e. The authors state that CML conditioned media resulted in a significant downregulation of BMDM phagocytosis, however this change is subtle and should be noted.

Response: As mentioned in our summary above and in Response to Reviewer 1, comments 1-3, we have significantly expanded this part. We now provide convincing evidence that treatment of mouse and human macrophages with CML conditional medium reduces phagocytosis, efferocytosis and CD36 expression (**new Figs. 3a-d; 5a-b, g-h, 6a-e; Suppl. Fig. S15a-b**). Similar effect is seen on efferocytosis following genetic or pharmacological inhibition of CD36, or lactotransferrin treatment, while the effect on phagocytosis occurs in CD36 independent way (**new Figs. 5a-b; 6a-f; Suppl. Fig. S15c**).

Relevant text has been added on Pages 12-16.

Line 278 / Figure 6e/f. The follow up on the secreted proteome is rather limited and does not completely explain the observed phenotypic differences in bone marrow macrophages. It is not clear why the authors did not evaluate other candidates identified in their screen.

We now provide much more detailed work on the effect lactotransferrin has on CD36 expression, macrophage function and maturation. We show that lactotransferrin supplementation reduces efferocytosis and phagocytosis of mouse and human macrophages, to similar degree as exposure to CML conditional media (**new Figs. 5a-b; 6a-c**), reduces CD36 levels in mouse and human macrophages (**new Figs. 5g-h; 6e**), increases levels of anti-inflammatory genes (**new Fig. 5g**) and reduces mitochondrial respiration in human macrophages (**new Suppl. Fig. S15d-e**). We also reveal that while the effect on efferocytosis is mediated by reduction in CD36, the effect on phagocytosis is CD36 independent (**new Figs. 6f; Suppl. Fig. S15c**). Therefore, while we agree that lactotransferrin is not the sole contributor to the observed phenotype, we provide strong evidence that it is an effective modulator of macrophage function. We also agree that the effect other secreted proteins have on macrophage function are of interest, but considering the new data, we believe it is now beyond the focus of our manuscript and should be followed up in separate projects.

Line 275 Related to the secreted proteome, the authors state the PCA revealed “clear separation between control and CML mice.” The supplied PCA shows minimal separation to my eye, suggesting either a small effect size or high degree of technical noise in this experiment. The authors should moderate their claims and or consider repeating the experiment to improve the quality.

Response: We have moderated the claim, now the text reads *“a noticeable separation between control and CML”*.

Line 125-126 “thus demonstrating that the CML niche supports myelopoiesis of the BCR-ABL negative population” This statement is not justified by the data. There is an expansion of BCR-ABL1 negative myeloid cells, but the authors do not provide any data supporting this being niche-derived.

Response: We acknowledge that we have not performed any specific experiments to prove this point and have therefore replaced the word *“demonstrating”* with *“suggesting”*.

Line 200-201 “Here we demonstrate that the CML BM niche supports an immune suppressive environment by driving high expression of *Tgfb1* and *Lgals1*.” This is an overstatement of the findings. It would be more correct to say that the CML BM niche results in decreased macrophage function which is associated high expression of *tgfb1* and *Lgals1*. The statement above suggests causation but data is not provided to support this.

Response: We acknowledge this and have moderated the claims (see also Response to Reviewer 2, comment 11)

New text on Page 11 reads: *“Overall, we demonstrate that the CML BM niche drives high expression of genes related to alternatively activated macrophages such as *Tgfb1* and *Lgals1*”*

Line 345-347 “Future work should aim to investigate if *cd36* expression can be used to monitor macrophage immune function and homeostasis in CML progression or during TKI treatment.” Murine and human cell surface markers often do not have 1:1 correspondence. In order for this to be a future direction, CD36 would have to mark this population in both mouse and human systems. Can the authors provide evidence to support this?

Response: As discussed above, we have confirmed that CD36 expression is reduced in CML exposed human macrophages and following lactotransferrin treatment (new Fig. 6d-e). However, to our best knowledge, there is no single cell transcriptomic data available on human Ph⁻ macrophages from CML patients following TKI treatment or in DMR. We hope that our work will ignite an interest in this area and highlight the importance of separating Ph⁻ CML exposed macrophages from Ph⁺ macrophages which are likely to be removed following TKI treatment in most chronic phase CML patients.

REVIEWER COMMENTS

Reviewer #2 mediating comments of reviewer #1

The manuscript presents experimental work using state-of-the-art methods studying the role of macrophages in CML on a single cell level. The authors used a chimeric CML mouse model to study the role CML cells on the marrow microenvironment and on local bystander macrophages. The major findings of the paper were:

- CML cells can alter macrophages transcriptomes and phagocytic function.
- CML exposed macrophages show down-regulation of phagocytosis genes CD36 and Lamp2.
- CML cells secrete factors including lactotransferrin, which suppress phagocytosis, mitochondrial respiration, and inflammatory genes.

Major concerns:

There are several major concerns with the manuscript:

1. All experiments were carried out with an (elegant) murine model or with murine cell lines. Therefore, the relevance of the findings for human disease remains unanswered. It would be highly desirable to conduct several experiments with human cells or tissues to create evidence for relevance in human CML.

Response: The authors have attempted to address this in three ways: a) The authors obtained primary human macrophages and performed in vitro culture experiments using conditioned medium from human CD34+ CML cells and normal HSC. They added an inhibitor of CD36 or lactotransferrin. They observed that this recapitulated the effects of conditioned medium from CML cells. b) They performed KO or KD studies of CD36 in THP1 macrophages, a human AML cell line. I would like to caution that this cell line is positive for the oncoprotein MLL-AF9, which will likely change macrophage biology. c) The authors analysed a scRNA-dataset showing that BM macrophages from CML patients are transcriptionally distinct and express an inflammatory signature. This is not surprising given that these macrophages, as the authors also point out, are BCR-ABL1+. But it is currently impossible to separate primary human BCR-ABL1+ versus BCR-ABL1- cells.

Conclusion: The authors did the best they could.

2. Overall, the manuscript remains descriptive in large parts. For example, no mechanistic

explanation is given on the regulatory molecules or signaling pathways that are induced in murine macrophages when exposed to CML cells or secreted factors (such as lactoferrin). No mechanistic explanation is given regarding the downregulation of some potentially relevant molecules such as CD36 or Lamp2.

Response: The authors added new data on the effect of lactotransferrin on efferocytosis and phagocytosis. However, the authors do not demonstrate how lactotransferrin is regulated. The authors did not find any involvement of Nf-KB signaling and did not attempt to find relevant signaling pathways.

Conclusion: The authors performed more mechanistic experiments, but could not identify the relevant signaling pathways in macrophages.

3. Similarly, on a functional level, the manuscript could benefit from functional experiments confirming the findings by alternative approaches. For example, the claim (e.g., Figure 3a) that the reduced expression of CD36 is functionally relevant for the phagocytic function of macrophages in the bone marrow of CML could be tested by blocking of CD36 in CD36 positive macrophages or CD36 gene deletion experiments. These experiments could be carried out, e.g., with coculture of CML cells or in functional tests of phagocytosis.

Response: The authors do deliver mechanistic experiments addressing this point.

Additional, major points:

Figure 2e: There is an apparent difference regarding the number of CD8+ T cells in the CML compartment versus the CML exposed compartment. How do the authors explain this difference?

Response: The authors provide a reasonable answer.

Figure 4a and b:

Can any of the clusters C1-C6 be found in human CML? It would be desirable to expose human macrophages to human CML cells (or supernatant) to investigate the relevance of the findings for human disease.

Response: The authors addressed this to the best of their abilities. All macrophages in this scRNA-set will likely be BCR-ABL1+ distorting the results. I agree that performing scRNA and DNA seq to test the inflammatory signature in BCR-ABL1 negative macrophages is beyond

the scope of this manuscript. The authors did perform in vitro experiments using human macrophages.

Figure 7a: could these experiments be reproduced with human macrophages or macrophage cell lines?

Response: This was done.

Figure 7g: In this figure, one can see significant changes only for TNFalpha and TGFbeta but not CD36, LAMP2 or others. This represents a contradiction to other experiments, e.g., of Figure 4d. This means that LTF only produces a subset of the changes seen with CML exposed cells. What other proteins of the secretome contribute to the observed effects?

Response: This experiment was repeated with a different experimental setup. The controversy has disappeared, and the authors acknowledge in their rebuttal letter that lactotransferrin is not the only contributing factor. The authors could be asked to add the fact, that lactotransferrin is not the only factor, to their discussion.

Figure 7h: The oxygen consumption rate shown upon 24-hr LTF stimulation drops somewhat, but this seems a rather small change. Please comment whether these changes are functionally relevant.

Response: The authors showed a similar reduction of the oxygen consumption rate after CD36 KO, CD36 inhibition and after lactotransferrin treatment, firming up these data. However, given the weak differences the authors moved the original data in figure 7h to the supplementary figures.

Minor comments

- Page 10, line 223 and other places: TNFalpha, please fill in the Greek symbol alpha. Done
- Figure 2: it would be easier to understand the legend of the figure if it described what is seen in the figure, e.g., to shortly explain the terms CML exposed versus CML. Moreover, there is a typo: "Ph- cells progenitor cells" in the title of the legend.

Response: This has been added to figure legend 1.

Reviewer #2 (Remarks to the Author):

This revised manuscript by Dawson et al. is significantly improved, although not all the suggested experiments were performed and definitive mechanistic experiments are still scarce. In particular, the flow of the main figures is improved. Few concerns remain:

1. The authors mention in their rebuttal letter that they included survival data after using a CSF1R antibody, but where are these data, in particular the survival curve?
2. Could the authors please expand on the reliability of using immunophenotypic markers to distinguish between Ph⁺ and Ph⁻ macrophages. This should be checked by FISH or a similar method.
3. Line 134: CD36 and Lamp2 have more functions than just phagocytosis.
4. Figure 3g: Can the authors show TNF α expression in CML-exposed macrophages?
5. What is the evidence that it is the exposure to BCR-ABL1⁺ LSC rather than mature BCR-ABL1⁺ neutrophils causing the observed effects?
6. THP1 cells are MLL-AF9⁺. This in itself could be influencing results and should at least be discussed.

Reviewer #3 (Remarks to the Author):

The authors have addressed the concerns raised in my point-by-point comments, and greatly improved the manuscript. However, it's still not clear whether macrophage function has any relationship to the clinical behavior of CML, or whether all the biology they describe is an epiphenomenon lacking clinical relevance. This was the my major point of concern in my initial review, and it remains unaddressed. The inclusion of the human single cell data does not really address this, as the authors acknowledge. Thus, the impact of this work remains moderate.

Summary: We would like to thank Reviewers 2 and 3 for their thorough and constructive comments. We would also like to thank Reviewer 2 for positively mediating the comments from Reviewer 1. Briefly, we have performed additional experiments in murine BM derived macrophages and human THP1 derived macrophages to address Reviewer 1 comments regarding macrophage signalling pathways affected by exposure to lactotransferrin. Here we show that 24h lactotransferrin exposure leads to decreased phosphorylation of the p65 subunit of NF- κ B, p38 MAPK and p44/42 MAPK (ERK1/2), which is in line with previous studies demonstrating that lactotransferrin has anti-inflammatory effect and reduces cytokine production (**new Supplementary Figure S14h**). Furthermore, similar effect was seen using human THP-1 derived macrophages (**new Supplementary Figure S15a**). Below we provide more detailed and point-by-point responses to all remaining comments.

Reviewer #2 mediating comments of reviewer #1

The manuscript presents experimental work using state-of-the-art methods studying the role of macrophages in CML on a single cell level. The authors used a chimeric CML mouse model to study the role CML cells on the marrow microenvironment and on local bystander macrophages. The major findings of the paper were:

- CML cells can alter macrophages transcriptomes and phagocytic function.
- CML exposed macrophages show down-regulation of phagocytosis genes CD36 and Lamp2.
- CML cells secrete factors including lactotransferrin, which suppress phagocytosis, mitochondrial respiration, and inflammatory genes.

Major concerns:

There are several major concerns with the manuscript:

1. All experiments were carried out with an (elegant) murine model or with murine cell lines. Therefore, the relevance of the findings for human disease remains unanswered. It would be highly desirable to conduct several experiments with human cells or tissues to create evidence for relevance in human CML.

Response: The authors have attempted to address this in three ways: a) The authors obtained primary human macrophages and performed in vitro culture experiments using conditioned medium from human CD34+ CML cells and normal HSC. They added an inhibitor of CD36 or lactotransferrin. They observed that this recapitulated the effects of conditioned medium from CML cells. b) They performed KO or KD studies of CD36 in THP1 macrophages, a human AML cell line. I would like to caution that this cell line is positive for the oncoprotein MLL-AF9, which will likely change macrophage biology. c) The authors analysed a scRNA-dataset showing that BM macrophages from CML patients are transcriptionally distinct and express an inflammatory signature. This is not surprising given that these macrophages, as the authors

also point out, are BCR-ABL1+. But it is currently impossible to separate primary human BCR-ABL1+ versus BCR-ABL1- cells.

Conclusion: The authors did the best they could.

Response: We thank Reviewer 2 for acknowledging our huge effort in successfully addressing this point.

2. Overall, the manuscript remains descriptive in large parts. For example, no mechanistic explanation is given on the regulatory molecules or signaling pathways that are induced in murine macrophages when exposed to CML cells or secreted factors (such as lactoferrin). No mechanistic explanation is given regarding the downregulation of some potentially relevant molecules such as CD36 or Lamp2.

Response: The authors added new data on the effect of lactotransferrin on efferocytosis and phagocytosis. However, the authors do not demonstrate how lactotransferrin is regulated. The authors did not find any involvement of Nf- κ B signaling and did not attempt to find relevant signaling pathways.

Conclusion: The authors performed more mechanistic experiments, but could not identify the relevant signaling pathways in macrophages.

Response: We have now performed additional experiments to address the Reviewers comment regarding “*signaling pathways that are induced in murine macrophages when exposed to CML cells or secreted factors (such as lactoferrin)*”. Despite the significant reduction in TNF α transcript levels following culture of murine BM derived macrophages in leukaemia condition medium, we (unexpectedly) did not observe clear changes in phosphorylation pattern of the p65 subunit of NF- κ B at this time-point (Supplementary Fig. S7d-e). However, to further investigate whether lactotransferrin affects NF- κ B signalling and other key signalling pathways, we performed additional experiments in murine BM derived macrophages and human THP1 derived macrophages. Interestingly, we show that 24h lactotransferrin exposure leads to decreased phosphorylation of the p65 subunit of NF- κ B, p38 MAPK and p44/42 MAPK (ERK1/2), which is in line with previous studies demonstrating that lactotransferrin has anti-inflammatory effect by suppressing cytokine production (new Supplementary Figure S14h**, and relevant reference added).**

Relevant text has been added on Page 15: “*Of note, we observed decreased phosphorylation of the p65 subunit of NF- κ B, p38 MAPK and p44/42 MAPK (ERK1/2) in BMDM exposed to LTF (Supplementary Fig. S14h). These results are in line with previous studies that have demonstrated that LTF has an anti-inflammatory effect by suppressing lipopolysaccharide (LPS)-induced cytokine production²³*”

Furthermore, we demonstrate that this is not limited to murine macrophages as lactotransferrin treatment had similar effect on human THP-1 derived macrophages (new Supplementary Figure S15a**), suggesting lactotransferrin may modulate key signalling downstream pathways.**

Relevant text has been added on Page 16: *“Additionally, LTF exposure decreased phosphorylation of the p65 subunit of NF-κB, p38 MAPK and ERK1/2 in control THP-1 macrophages in both the presence and absence of LPS (Supplementary Fig. S15a).”*

3. Similarly, on a functional level, the manuscript could benefit from functional experiments confirming the findings by alternative approaches. For example, the claim (e.g., Figure 3a) that the reduced expression of CD36 is functionally relevant for the phagocytic function of macrophages in the bone marrow of CML could be tested by blocking of CD36 in CD36 positive macrophages or CD36 gene deletion experiments. These experiments could be carried out, e.g., with coculture of CML cells or in functional tests of phagocytosis.
Response: The authors do deliver mechanistic experiments addressing this point.

Response: We thank Reviewer 2 for acknowledging our effort in addressing this point.

Additional, major points:

Figure 2e: There is an apparent difference regarding the number of CD8+ T cells in the CML compartment versus the CML exposed compartment. How do the authors explain this difference?

Response: The authors provide a reasonable answer.

Response: This has now been addressed.

Figure 4a and b:

Can any of the clusters C1-C6 be found in human CML? It would be desirable to expose human macrophages to human CML cells (or supernatant) to investigate the relevance of the findings for human disease.

Response: The authors addressed this to the best of their abilities. All macrophages in this scRNA-set will likely be BCR-ABL1+ distorting the results. I agree that performing scRNA and DNA seq to test the inflammatory signature in BCR-ABL1 negative macrophages is beyond the scope of this manuscript. The authors did perform in vitro experiments using human macrophages.

Response: This has now been addressed.

Figure 7a: could these experiments be reproduced with human macrophages or macrophage cell lines?

Response: This was done.

Response: This has now been addressed.

Figure 7g: In this figure, one can see significant changes only for TNFalpha and TGFbeta but not CD36, LAMP2 or others. This represents a contradiction to other experiments, e.g., of Figure 4d. This means that LTF only produces a subset of the changes seen with CML exposed cells. What other proteins of the secretome contribute to the observed effects?

Response: This experiment was repeated with a different experimental setup. The controversy

has disappeared, and the authors acknowledge in their rebuttal letter that lactotransferrin is not the only contributing factor. The authors could be asked to add the fact, that lactotransferrin is not the only factor, to their discussion.

Response: We agree and have added relevant sentence to the Discussion Page 20: “It should however be noted that LTF is likely one of many factors contributing to the observed cellular phenotype in this intricate environment of cytokines, chemokines, secreted immunomodulatory proteins and the complex array of cellular interactions.”

Figure 7h: The oxygen consumption rate shown upon 24-hr LTF stimulation drops somewhat, but this seems a rather small change. Please comment whether these changes are functionally relevant.

Response: The authors showed a similar reduction of the oxygen consumption rate after CD36 KO, CD36 inhibition and after lactotransferrin treatment, firming up these data. However, given the weak differences the authors moved the original data in figure 7h to the supplementary figures.

Response: This has now been addressed.

Minor comments

- Page 10, line 223 and other places: TNFalpha, please fill in the Greek symbol alpha. Done
- Figure 2: it would be easier to understand the legend of the figure if it described what is seen in the figure, e.g., to shortly explain the terms CML exposed versus CML. Moreover, there is a typo: “Ph- cells progenitor cells” in the title of the legend.

Response: This has been added to figure legend 1.

Response: This has now been addressed.

Reviewer #2 (Remarks to the Author):

This revised manuscript by Dawson et al. is significantly improved, although not all the suggested experiments were performed and definitive mechanistic experiments are still scarce. In particular, the flow of the main figures is improved. Few concerns remain: 1. The authors mention in their rebuttal letter that they included survival data after using a CSF1R antibody, but where are these data, in particular the survival curve?

Response: We thank Reviewer 2 acknowledging that our Revised paper has significantly improved. Data/survival curve is now presented in Figure 2c.

2. Could the authors please expand on the reliability of using immunophenotypic markers to distinguish between Ph+ and Ph- macrophages. This should be checked by FISH or a similar method.

Response: In Supplementary Figure S4c we demonstrated that BCR-ABL expression is not induced in the CD45.1 CD11b+F4/80+ macrophages as expected. Given that the human BCR-

ABL gene (not the Philadelphia chromosome) is used in this mouse model, FISH (detects fusion of two chromosomes) is not an appropriate assay to check for BCR-ABL+ macrophages in the mouse model.

3. Line 134: CD36 and Lamp2 have more functions than just phagocytosis.

Response: We agree with this point and have added relevant sentence on Page 8: “*Both Cd36 and Lamp2 are known to be associated with many functions, including, to our interest, phagocytosis.*”

4. Figure 3g: Can the authors show TNFalpha expression in CML-exposed macrophages?

Response: In Figure 5g we show TNF α expression in CML exposed BM derived macrophages (+/- LFT treatment; and in Supplementary. Fig. S14g, in normal medium). In Supplementary Fig. S7d TNF α expression is shown in BM derived macrophages following culture in leukaemia conditional medium.

5. What is the evidence that it is the exposure to BCR-ABL1+ LSC rather than mature BCR-ABL1+ neutrophils causing the observed effects?

Response: We agree and cannot exclude that other cell types secrete lactotransferrin, or any of the other molecules that mediated the effect. Given that it is the primitive LSC population that persists in CML patients, we focussed on this cell populations in this work.

6. THP1 cells are MLL-AF9+. This in itself could be influencing results and should at least be discussed.

Response: We agree with this point and have added relevant sentence in Discussion Pages 20-21: “*It will be appreciated that while THP-1 cells are a model of human macrophages in vitro, THP-1 cells are MLL-AF9+. Expression of the MLL-AF9+ oncogene may be a contributing factor to the phenotype observed in this study and, as such, the results should be interpreted with this in mind. However, in an attempt to address this shortcoming, we were able to demonstrate that macrophages derived from normal human BM also displayed significantly reduced efferocytosis.*”

Reviewer #3 (Remarks to the Author):

The authors have addressed the concerns raised in my point-by-point comments, and greatly improved the manuscript. However, its still not clear whether macrophage function has any relationship to the clinical behavior of CML, or whether all the biology they describe is an epiphenomenon lacking clinical relevance. This was the my major point of concern in my initial review, and it remains unaddressed. The inclusion of the human single cell data does not really address this, as the authors acknowledge. Thus, the impact of this work remains moderate.

Response: We thank Reviewer 3 for positive feedback and for acknowledging our efforts in improving the manuscript. We agree that further investigation is required to deliver clinical impact but anticipate that our manuscript, which describes changes in CML-exposed

macrophage function for the first time, may ignite further investment into this area of research. Indeed, we have already received additional funding to investigate whether therapeutic antibodies can be used to enhance macrophage effector function and phagocytosis of leukaemic cells in the bone marrow niche, and we are aware of other groups expanding their analysis of bystander immune cells into the macrophage population.

REVIEWERS' COMMENTS

Reviewer #2 (Remarks to the Author):

The authors have done an excellent job at addressing the concerns of reviewers #1 and #2. The fact that it is unclear whether the LSC or more mature CML cells are responsible for the observed effects could be added to the discussion.

REVIEWERS' COMMENTS

Reviewer #2 (Remarks to the Author):

The authors have done an excellent job at addressing the concerns of reviewers #1 and #2. The fact that it is unclear whether the LSC or more mature CML cells are responsible for the observed effects could be added to the discussion.

Response: We are pleased that our revisions have addressed all the reviewers' comments and would like to thank Reviewer 2 for their excellent comments and feedback. We have added relevant sentence to the Discussion section: *“Furthermore, neutrophil granules have been found to contain LTF³⁸, thus we cannot exclude that mature cells in the bone marrow niche secrete LTF in a similar manner to primitive CML cells.”*